# Mechanism of Ψ-Pro/C-degron recognition by the CRL2^FEM1B ubiquitin ligase

Xinyan Chen[1,2,6], Anat Raiff[3,6], Shanshan Li ®[1], Qiong Guo[1], Jiahai Zhang[1], Hualin Zhou[1], Richard T. Timms ®[4], Xuebiao Yao ®[1], Stephen J. Elledge ®[5], Itay Koren ®[3] ✉, Kaiming Zhang ®[1] ✉ & Chao Xu ®[1,2] ✉

The E3 ligase-degron interaction determines the specificity of the ubiquitin–proteasome system. We recently discovered that FEM1B, a substrate receptor of Cullin 2-RING ligase (CRL2), recognizes C-degrons containing a C-terminal proline. By solving several cryo-EM structures of CRL2^FEM1B bound to different C-degrons, we elucidate the dimeric assembly of the complex. Furthermore, we reveal distinct dimerization states of unmodified and neddylated CRL2^FEM1B to uncover the NEDD8-mediated activation mechanism of CRL2^FEM1B. Our research also indicates that, FEM1B utilizes a bipartite mechanism to recognize both the C-terminal proline and an upstream aromatic residue within the substrate. These structural findings, complemented by in vitro ubiquitination and in vivo cell-based assays, demonstrate that CRL2^FEM1B-mediated polyubiquitination and subsequent protein turnover depend on both FEM1B-degron interactions and the dimerization state of the E3 ligase complex. Overall, this study deepens our molecular understanding of how Cullin-RING E3 ligase substrate selection mediates protein turnover.

The ubiquitin–proteasome system (UPS) is the major route for the orchestration of proteostasis by clearing misfolded or unwanted proteins in eukaryotes[1,2]. Deficiency in the UPS leads to severe human diseases, including cancers and neurodegenerative diseases[3,4]. Typically, polyubiquitination (PolyUb), a process by which ubiquitin (Ub) molecules are conjugated to substrate proteins, serves as a targeting signal for destruction of the substrate by the proteasome[1,5]. PolyUb is coordinated by ubiquitin-activating enzymes (E1), ubiquitin-conjugating enzymes (E2), and ubiquitin ligases (E3)[6]. In most cases, E3 directly binds the substrate to mediate the transfer of ubiquitin from the E2-Ub thioester intermediate to a specific lysine of the substrate[7].

More than 600 E3s are encoded in the human genome, of which Cullin-RING E3 ligases (CRLs) are the largest family. CRLs are multisubunit ubiquitin ligases, with a Cullin protein serving as the scaffold. Activation of the CRL complex requires neddylation, the conjugation of the ubiquitin-like protein NEDD8 to the cullin protein[8–10]. There are eight Cullin members encoded in the human genome (CUL1, 2, 3, 4A, 4B, 5, 7, and 9)[11]. The N-terminal helical region and C-terminal WHB domain of Cullin interact with an adapter protein and a RING domain protein (RBX1 or RBX2), respectively[9]. In some CRLs, such as CRL3, the adapter protein also mediates substrate binding[12], while in others, the adapter protein further associates with an interchangeable subunit, termed the substrate receptor (SR), for substrate recognition[13]. A specific CRL is characterized by the Cullin protein, and the SR is denoted with a superscript, such as CRL2^VHL, indicating the Cullin 2 (CUL2) E3 with VHL as the SR[14].

[1]MOE Key Laboratory for Cellular Dynamics, Division of Life Sciences and Medicine, University of Science and Technology of China, Hefei 230027, PR China. [2]Center for Advanced Interdisciplinary Science and Biomedicine of IHM, Division of Life Sciences and Medicine, University of Science and Technology of China, Hefei 230027, PR China. [3]The Mina and Everard Goodman Faculty of Life Sciences, Bar-Ilan University, Ramat-Gan 5290002, Israel. [4]Cambridge Institute of Therapeutic Immunology and Infectious Disease, Department of Medicine, University of Cambridge, Cambridge, UK. [5]Division of Genetics, Department of Medicine, Howard Hughes Medical Institute, Brigham and Women's Hospital, Harvard Medical School, Boston, MA 02115, USA. [6]These authors contributed equally: Xinyan Chen, Anat Raiff. ✉e-mail: itay.koren@biu.ac.il; kmzhang@ustc.edu.cn; xuchao@ustc.edu.cn

It has been recently reported that dedicated CRLs target specific protein sequences located at the N-termini and C-termini of proteins, termed N- and C-degrons[15], to mediate substrate turnover. A degron is a transferable destabilizing signal that can recruit an E3 to mediate PolyUb and subsequent proteasomal proteolysis of the substrate[16]. The N-degron and C-degron pathways play important roles in a wide spectrum of cellular functions, including the cell cycle, DNA repair, and stress responses[15,17]. Previously, utilizing the genome-wide Global Protein Stability (GPS) assay, we and others reported that CRL2[APPBP2], CRL2[FEM1A/C], CRL2[FEM1B], CRL2[KLHDC2/3/10], CRL4[TRPC4AP] and CRL4[DCAF12] serve as C-degron-binding E3s[17–20]. Subsequent structural studies revealed the underlying C-degron recognition mode of various SRs, including KLHDC2[21,22], FEM1B[20], DCAF12[23], and APPBP2[19].

FEM1B, an SR for a CRL2 E3 complex, is of special interest since we and others found that it is able to recognize the distinct substrates CDK5R1 and FNIP1 via the Arg/C-degron and the zinc-binding motif, respectively[20,24]. In addition, FEM1B demonstrates different substrate-binding properties compared with its two close homologs, FEM1A and FEM1C[20]. Very recently, in a systematic screen to identify CRL substrates, CRL2[FEM1B] was suggested to recognize several additional substrates bearing C-degrons, including those of CCDC89, CUX1, and PSMB5[25,26]. Intriguingly, none of these C-degrons contained a previously known FEM1B-binding motif; instead, most FEM1B substrates terminated with proline (Pro) were identified. However, how CRL2[FEM1B] is assembled and recognizes these identified Pro/C-degrons is largely unknown.

Here, to unveil the molecular mechanism underlying C-degron recognition, we employ cryogenic electron microscopy (cryo-EM) to determine the structures of unmodified or neddylated CRL2[FEM1B] bound with the C-degron of CCDC89. Unexpectedly, unmodified CRL2[FEM1B] adopts two different dimerization states, with one symmetric and the other asymmetric. Upon neddylation, the asymmetric dimer is converted to a third dimerization state that likely represents the active form. Structural analysis reveals that FEM1B utilizes the Arg/C-degron binding site and a previously unidentified hydrophobic pocket to coordinately bind long, linear C-degrons composed of a C-terminal Pro and an aromatic reside located ~20 residues upstream. Biochemistry experiments and cellular protein stability assays demonstrated that disruption of CRL2[FEM1B] dimerization or interaction between FEM1B and C-degron impairs the substrate PolyUb in vitro and ubiquitin-dependent protein turnover in vivo. This study not only provides structural insights into the recently identified C-degron pathway mediated by the CRL2[FEM1B] dimer but also uncovers another potential druggable pocket of FEM1B for designing proteolysis-targeting chimeras (PROTACs) and molecular glues for targeted protein degradation (TPD).

## Results

### Structure of the symmetric dimer of unmodified CRL2[FEM1B]

To provide structural insight into the molecular mechanism underlying C-degron recognition by full-length FEM1B (FEM1B[FL]), we purified the human FEM1B-elongin B (EB)-elongin C (EC) complex, as well as the CUL2Δ-RBX1 dimer. CUL2Δ is a variant of CUL2 with an intrinsically disordered region removed to prevent unwanted proteolysis that has been reported to retain its activity similar to that of its wild-type (WT) counterpart[19,27]. Hereafter, CUL2Δ reconstituted in the E3 complex was named CUL2 for simplicity. The FEM1B-EB-EC and CUL2-RBX1 complexes were mixed at a 1:1 ratio and further purified by gel filtration to obtain the CRL2[FEM1B] quinary complex (Fig. 1a, Supplementary Fig. 1a).

An excess of the CCDC89 C-degron peptide was added to obtain the cryo-EM sample of the CRL2[FEM1B]-CCDC89 complex. Following data acquisition and processing, our results indicate the presence of two distinct dimerization states for the particles. One state exhibits a symmetrical configuration, while the other adopts an asymmetrical conformation. The cryo-EM structures of the symmetric and

asymmetric dimers of the CRL2[FEM1B] heteropentamer (FEM1B-EB-EC-CUL2-RBX1) were solved at resolutions of 3.39 Å and 3.37 Å, respectively (Fig. 1b–e, Supplementary Fig. 2 and Supplementary Table 1). The maps of the two structures are of high quality, allowing us to build atomic models for most regions of the CRL2[FEM1B] subunits. Hereafter, the symmetric dimer is referred to as dimer[s] (Fig. 1b, c), and the asymmetric dimer is designated as dimer[a] (Fig. 1d, e). The unmodified CRL2[FEM1B] is referred to as [un]CRL2[FEM1B].

In the dimer[s] structure, most residues in FEM1B, EB, EC, CUL2, and RBX1 are visible (Fig. 2a, b). Given that the structures of both CRL2[Lrr1] and CRL2[VHL] behave as monomers[14,28,29], whereas CRL2[KLHDC2] assembles as a tetramer[22], we hypothesized that the dimerization of two CRL2[FEM1B] protomers (CRL2.p1 and CRL2.p2) is determined by the SR FEM1B. FEM1B adopts a V-shaped architecture consisting of seven N-terminal ankyrin repeats (ANK1-7), five TPR repeats (TPR1-5), and three C-terminal ankyrin repeats (ANK8-10) followed by a putative VHL box (VB) (Fig. 1a, Supplementary Fig. 3).

In the dimer[s], the protomer-protomer interface is symmetric. Specifically, the two FEM1B molecules, named as FEM1B.p1 and FEM1B.p2, interact with each other via a hydrophobic patch formed by two C-terminal ankyrin repeats (ANK9-10), consisting of Phe549 and His553 in ANK9 and Val584, Ile587, and Leu588 in ANK10 (Fig. 2a, c), which are conserved in FEM1B orthologs but not in FEM1A and FEM1C (Supplementary Fig. 3)[20]. Two head-to-tail juxtaposed CUL2-RBX1 dimers wrap around the FEM1B homodimer interface to stabilize the architecture (Fig. 1b, c, Supplementary Fig. 4a).

In each CRL2[FEM1B] protomer, one FEM1B molecule interacts with an EB-EC dimer and a CUL2 molecule via the VB at the extreme C-terminus (Fig. 1b, c, Supplementary Fig. 4b). The overall architectures of the two CRL2[FEM1B] protomers in the dimer[s] are almost identical, as evidenced by a root-mean-square deviation (RMSD) of 1.66 Å over 1600 main chain Cα atoms (Supplementary Fig. 4b). The VB of FEM1B, which encompasses α32-α33, adopts a helix-turn-helix conformation (Supplementary Fig. 4b) and interacts with EC and CUL2 through hydrophobic interactions (Fig. 2d). Specifically, Leu597 and Leu600 of FEM1B are buried into a large hydrophobic groove of EC formed by Val73, Tyr76, Phe77, Phe93, Ile95, and Leu103; Leu104 of EC is accommodated into a hydrophobic pocket of FEM1B composed of Leu620, Phe623, and Val624. Additional hydrophobic interactions were found between FEM1B Pro617 and CUL2 Pro5 and Val47 (Fig. 2d).

It has been reported that the WHB domain of CUL2 interacts with RBX1, resulting in an inactive form of E3[30]. This complex can only be activated through CUL2 neddylation, which disrupts the WHB-RBX1 interaction and triggers the release of RBX1, allowing E3 to function[31]. Unexpectedly, in one CRL2[FEM1B] protomer (CRL2.p2), the WHB domain of CUL2.p2 (WHB.p2) does not directly contact RBX1.p2; instead, both CUL2.p2 and RBX1.p2 interact with the N-terminal region of CUL2 from the other protomer (CUL2.p1), thereby forming the CUL2.p2-CUL2.p1-RBX1.p1 interprotomer interaction (Fig. 2e, f). Val7, Tyr49, and Pro50 of CUL2.p1 makes hydrophobic interactions with Ile44, Arg86, and Trp87 of RBX1.p2; an additional cation-π interaction is found between RBX1.p2 Arg91 and CUL2.p1 Tyr49 (Fig. 2e); and CUL2.p2 loosely associates with CUL2.p1 via electrostatic interactions (Fig. 2f).

### Structure of the asymmetric dimer of unmodified CRL2[FEM1B]

The FEM1B-CUL2-EC interactions within a CRL2[FEM1B] protomer are conserved in both structures of dimer[s] and dimer[a] (Fig. 2d). However, the overall architecture and the FEM1B.p1-FEM1B.p2 interfaces are different for the symmetric and asymmetric dimers. Unlike dimer[s], dimer[a] adopts a ring-like architecture, with the two CRL2 protomers binding to each other in an asymmetric manner (Fig. 1d, e, Supplementary Figs. 4c and 5a). Specifically, the C-terminal hydrophobic patch of FEM1B.p2 interacts with α17 of FEM1B.p1 (Supplementary Fig. 5b). The C-terminal hydrophobic patch of FEM1B.p1 in turn associates with RBX1.p2 to seal the ring, with Ile546, Phe549, and

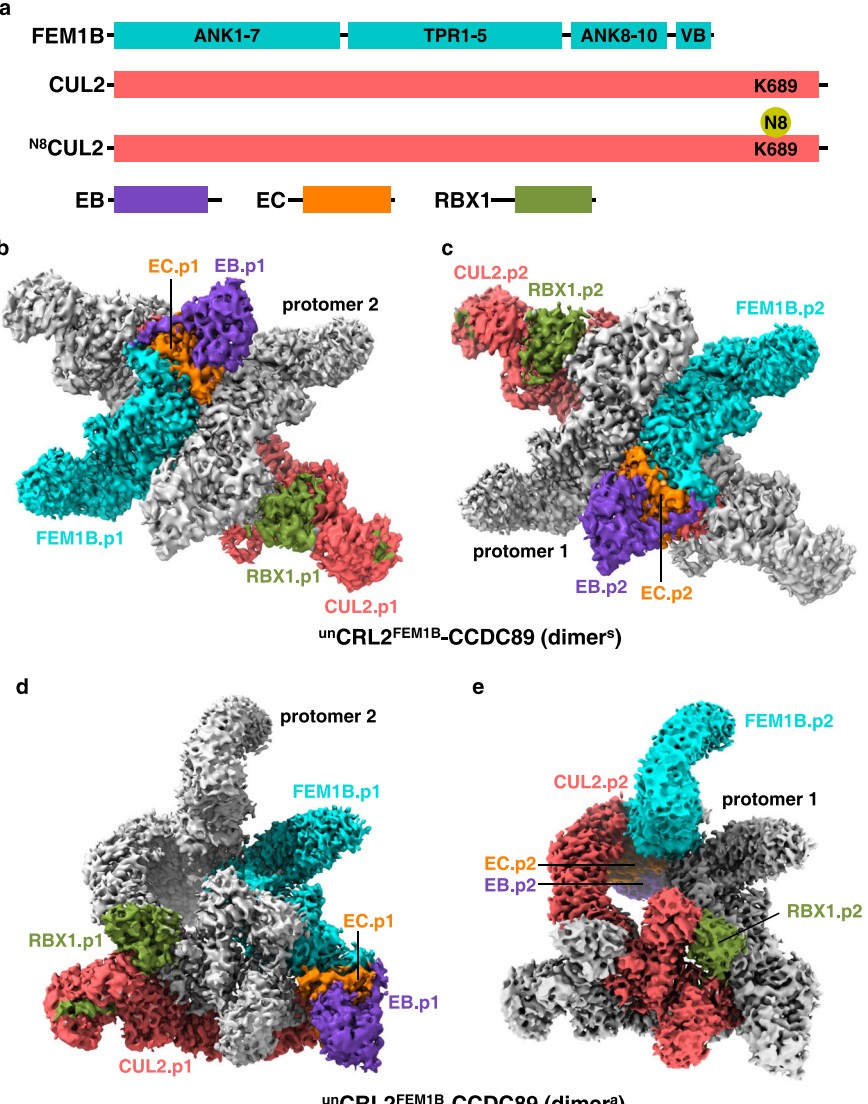

**Fig. 1 | Overall structure of $^{un}$CRL2$^{FEM1B}$. a** Domain architecture of individual subunits in the CRL2$^{FEM1B}$ complex, with each subunit denoted by a distinct color. Maps of the $^{un}$CRL2$^{FEM1B}$ dimer$^s$ bound to the CCDC89 C-degron, with protomer 1 colored by subunits and protomer 2 colored gray (**b**), or vice versa (**c**). Maps of the $^{un}$CRL2$^{FEM1B}$ dimer$^a$ bound to the CCDC89 C-degron, with protomer 1 colored by subunits and protomer 2 in gray (**d**), or vice versa (**e**).

Leu550 of FEM1B.p1 forming hydrophobic interactions with Trp35, Ile37, Ile49, Met50, and Trp72 of RBX1.p2 (Supplementary Fig. 5c). RBX1.p2 associates with the WHB domain of CUL2.p2, which characterizes an autoinhibitory state for unmodified CRL2. In contrast to RBX1.p2, RBX1.p1 does not contact the WHB domain of CUL2.p1 but associates with the middle region of CUL2.p2, providing additional interprotomer interactions (Supplementary Fig. 4d). CUL2.p2 Arg325 interacts with Trp35 and Leu96 of RBX1.p1 via cation-π and hydrogen bonding interactions, respectively; CUL2.p2 Tyr378 interacts with Arg99 and Glu100 of RBX1.p1 via cation-π and hydrogen bonding interactions, respectively; and RBX1.p1 Arg99 also forms hydrogen bonds with Thr327 and Glu380 of CUL2.p2 (Supplementary Fig. 5d).

In the dimer$^a$ structure, the CUL2.p2 WHB domain interacts with RBX1.p2, with Lys689 forming electrostatic interactions with RBX1.p2 Glu55 (Supplementary Fig. 5e); the CUL2.p1 WHB domain associates with CUL2.p2, with Lys689 forming hydrogen bonds with Gly360 and Gln362 of CUL2.p2 (Supplementary Fig. 5f). Therefore, Lys689 neddylation disrupts the assembly of the dimer$^a$ by introducing potential steric clashes.

To understand the structural rearrangement in the asymmetric assembly of dimer$^a$, we superimposed the two CRL2$^{FEM1B}$ protomers on CUL2 and found that while EB and EC could be superimposed very well, FEM1B.p1 shifted 19 Å, and RBX1.p1 rotated ~150° (Supplementary Fig. 4d). Taken together, our findings demonstrate that ANK9-10 plays a key role in mediating the asymmetric dimerization of FEM1B within dimer$^a$. Furthermore, the observed subunit rearrangements endow CRL2$^{FEM1B}$ with the ability to adopt two distinct dimerization modes. Interestingly, the two RBX1 molecules exhibit different orientations, likely to accommodate these varied assembly states.

## Structure of neddylated CRL2$^{FEM1B}$

It has been reported that Cullin neddylation induces conformational changes that activate CRLs by releasing the autoinhibited conformation of E3[31]. Consistently, in both unmodified CRL2$^{FEM1B}$ ($^{un}$CRL2$^{FEM1B}$) structures, the CUL2 WHB domains are all visible, and the potential lysine neddylation site (Lys689) is partially buried, suggesting that CUL2 Lys689 neddylation is compatible with neither dimer. To understand how CUL2 Lys689 neddylation affects the structure and activity of CRL2$^{FEM1B}$, we further reconstituted the neddylated CRL2$^{FEM1B}$ ($^{N8}$CRL2$^{FEM1B}$) complex bound to the CCDC89 C-degron, in which Lys689 of CUL2 is modified by NEDD8 (Supplementary Fig. 1b, c).

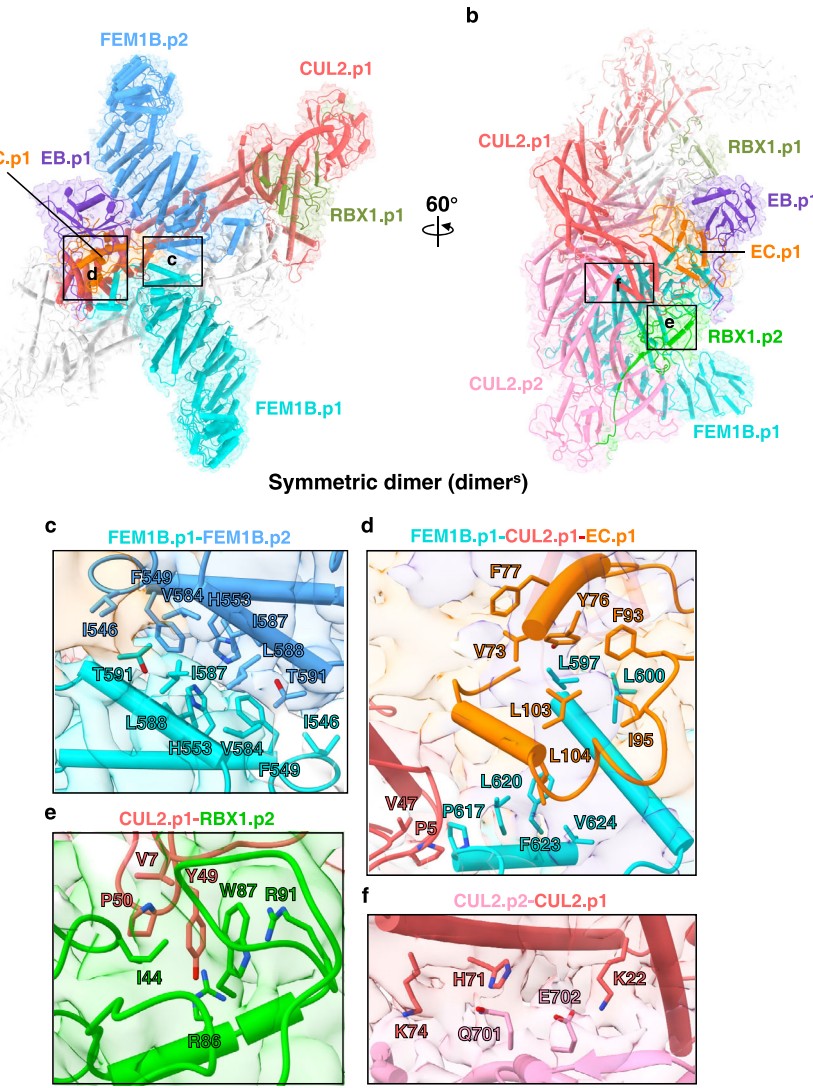

**Fig. 2 | Interaction network within the $^{un}$CRL2$^{FEM1B}$ dimer$^s$. a, b** Overall structure and cryo-EM map of the unmodified CRL2$^{FEM1B}$ dimer$^s$, with the subunits shown in cartoon representation. The subunits of protomer 1 are colored the same as those in Fig. 1a. Close-up views of the FEM1B.p1-FEM1B.p2 interface (**c**), FEM1B.p1-CUL2.p1-EC.p1 interface (**d**), CUL2.p1-RBX1.p2 interface (**e**), and CUL2.p2-CUL2.p1 interface (**f**). The interaction residues are shown in the context of the cryo-EM map.

Although $^{N8}$CRL2$^{FEM1B}$ still behaves as a dimer, the $^{N8}$CRL2$^{FEM1B}$ particles display two dimerization states. While the architecture of one dimerization state (state 1) is similar to that of dimer$^s$ (Fig. 3a, b), the architecture of the other (state 2) is distinct from that observed in $^{un}$CRL2$^{FEM1B}$ (Fig. 3c, d). The two dimerization states of the $^{N8}$CRL2$^{FEM1B}$-CCDC89 complex, named dimer$^{S1}$ and dimer$^{S2}$, were solved by cryo-EM at resolutions of 3.44 Å and 4.09 Å, respectively (Supplementary Fig. 6 and Supplementary Table 1).

The overall architecture of dimer$^{S1}$ of $^{N8}$CRL2$^{FEM1B}$ is similar to that of $^{un}$CRL2$^{FEM1B}$ dimer$^s$ except that the WHB domains of both CUL2 are invisible, suggesting their flexible orientation (Figs. 1b, c and 3a, b). On the other hand, the dimer$^a$ of $^{un}$CRL2$^{FEM1B}$ is disrupted by CUL2 neddylation (Supplementary Fig. 5e, f). Conversely, the installed NEDD8 proteins induce a distinct symmetric dimerization state of $^{N8}$CRL2$^{FEM1B}$, named dimer$^{S2}$ (Fig. 3c, d).

To obtain a higher-resolution structure to unveil the dimer$^{S2}$ assembly, we collected another cryo-EM dataset for the $^{N8}$CRL2$^{FEM1B}$ complex bound to the CDK5R1 C-degron and solved the dimer$^{S2}$ structure of $^{N8}$CRL2$^{FEM1B}$-CDK5R1 at an overall resolution of 3.54 Å (Supplementary Fig. 7 and Supplementary Table 1). Given that the overall architectures of both dimer$^{S2}$ structures are exactly the same (Fig. 3c–f), we used the higher resolution dimer$^{S2}$ structure of

$^{N8}$CRL2$^{FEM1B}$-CDK5R1 for structural analysis. The dimer$^{S2}$ is symmetric and mediated by two NEDD8 molecules (Supplementary Fig. 8a), which display weaker density than the other complex subunits, allowing us to build only their main chains. Each NEDD8 molecule is located close to the FEM1B-CUL2 interface and likely forms hydrophobic interactions with the C-terminal hydrophobic patch of FEM1B (ANK9-10) via Ile44 (Supplementary Fig. 8b). In addition, the two RBX1 molecules, RBX1.p1 and RBX1.p2, interact with each other via symmetric hydrophobic interactions, with Trp35 and Ile37 from one RBX1 contacting Ile44, Trp87, and Val93 of the other (Supplementary Fig. 8c).

Given that the hydrophobic patch of FEM1B is involved in dimerization assembly in all solved structures, we proposed that mutation of the hydrophobic patch residues would affect the dimerization of CRL2$^{FEM1B}$. To this end, we reconstituted the $^{N8}$CRL2$^{FEM1B}$ complex with the FEM1B F549D/V584D/I587D/L588D mutant (4D mutant) (Supplementary Fig. 1d). Consistent with the structural analysis, the static light scattering (SLS) experiment shows that the molecular weight of $^{N8}$CRL2$^{FEM1B}$ is 325 kDa, indicating that it is a dimer in solution, while the $^{N8}$CRL2$^{FEM1B}$ 4D mutant behaves as a monomer (molecular weight (MW): 171 kDa) (Supplementary Fig. 1e). Taken together, the C-terminal hydrophobic patch of FEM1B is critical for dimerization of the $^{N8}$CRL2$^{FEM1B}$ complex.

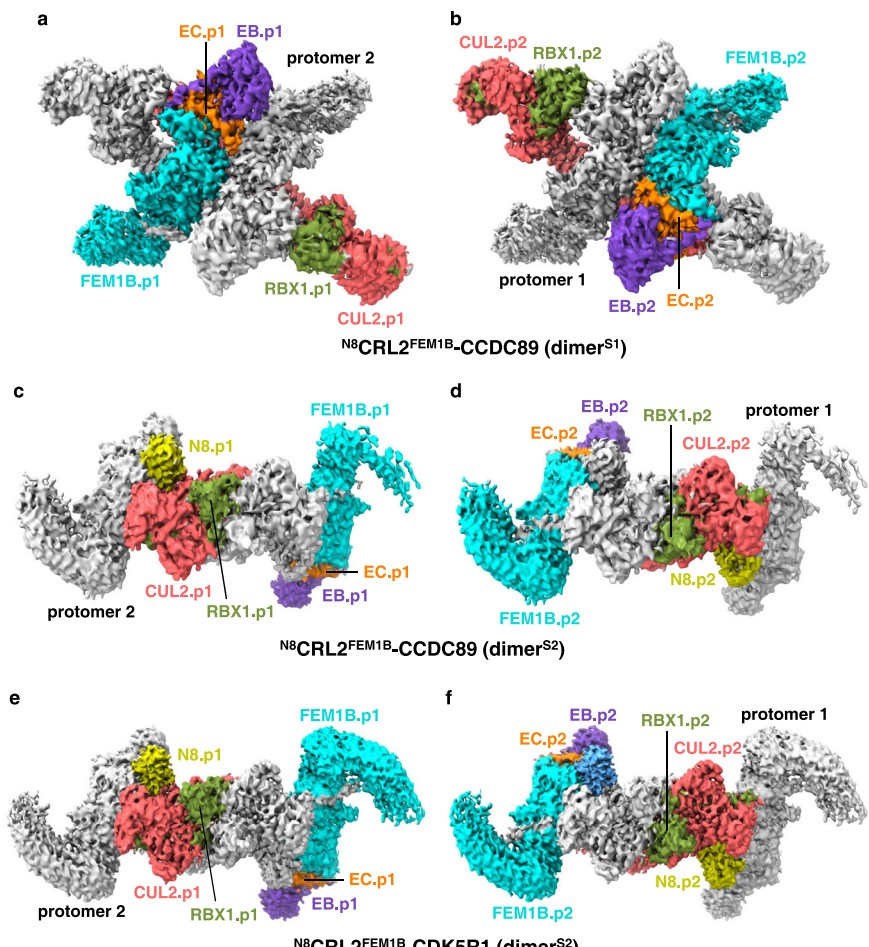

**Fig. 3 | Overall structure of $^{N8}$CRL2$^{FEM1B}$. a, b** Maps of the $^{N8}$CRL2$^{FEM1B}$ dimer$^{S1}$ bound to the CCDC89 C-degron, with protomer 1 colored by subunits and protomer 2 colored gray (**a**), or vice versa (**b**). **c, d** Maps of the $^{N8}$CRL2$^{FEM1B}$ dimer$^{S2}$ bound to the CCDC89 C-degron, with protomer 1 colored by subunits and protomer 2 colored gray (**c**), or vice versa (**d**). Maps of the $^{N8}$CRL2$^{FEM1B}$ dimer$^{S2}$ bound to the CDK5R1 C-degron, with protomer 1 colored by subunits and protomer 2 colored gray (**e**), or vice versa (**f**).

## CRL2$^{FEM1B}$ assembles into a dimeric state in vivo

To investigate the oligomeric state of CRL2$^{FEM1B}$ in vivo, we employed gel filtration chromatography on lysates extracted from cells stably expressing either WT FEM1B or the 4D mutant. In line with the structural data, the WT complex is eluted in higher MW fractions across all collected samples than the complex containing the FEM1B 4D mutant, suggesting that the WT protein forms oligomers in vivo, whereas the 4D mutation impairs this oligomerization process (Supplementary Fig. 9a, b). Next we used co-immunoprecipitation to further corroborate the in vivo dimerization of FEM1B. Cells co-expressing HA- and MYC-tagged WT or 4D mutant FEM1B were subjected to immunoprecipitation using an anti-HA antibody. Compared with those of the 4D mutant, significantly greater amounts of WT MYC-FEM1B co-precipitated with WT HA-FEM1B (Supplementary Fig. 9c), supporting the notion of enhanced dimerization between WT FEM1B proteins compared to dimerization-deficient mutant in vivo. Furthermore, there was greater co-precipitation of endogenous CUL2 from immunoprecipitated WT HA-FEM1B compared to the 4D mutant (Supplementary Fig. 9d), implying that FEM1B dimerization likely facilitates binding to CUL2 and stabilizes the CRL2$^{FEM1B}$ complex.

## The autoinhibitory state of RBX1 is partially released by CUL2 neddylation

To gain mechanistic insight into how NEDD8 promotes the activation of CRL2$^{FEM1B}$, we first superimposed the structure of RBX1-UBE2D2-Ub (PDB: 6TTU) onto the structure of $^{un}$CRL2$^{FEM1B}$ dimer$^s$ bound to RBX1, the RING protein. There are potential steric clashes between the E2 enzyme (UBE2D2) and CUL2.p2 , and between Ub and the WHB domain of CUL2.p1 (Supplementary Fig. 10a), demonstrating that the CUL2.p2-RBX1.p1 interaction is incompatible with the RBX1-UBE2D2-Ub complex. The interface between RBX1 and UBE2D2-Ub is much larger than that between RBX1.p1 and CUL2.p2 (1030 vs. 420 Å$^2$), implying that UBE2D2-Ub competes for CUL2.p2 upon binding to RBX1.p1.

Next, we superimposed the structures of RBX1-UBE2D2-Ub and $^{N8}$CRL2$^{FEM1B}$ dimer$^{S1}$ onto RBX1 (Supplementary Fig. 10b). The structure superposition indicates that CUL2 neddylation leads to the disassociation of the CUL2.p1 WHB from CUL2.p2, leaving more space for Ub to contact RBX1.p1, which facilitates stronger competition of UBE2D2-Ub with CUL2.p2 for binding to RBX1.p1. CUL2 neddylation likely alleviates the inhibitory state of dimer$^s$ by leading to the release of the CUL2.p1 WHB domain from CUL2.p2, which decreases the energy barrier for the interaction of UBE2D2-Ub with RBX1.

## FEM1B binds to long C-degrons ending with proline

Recent work reported that CRL2$^{FEM1B}$ E3 binds to the C-degrons of CCDC89, PSMB5, BEX2, CUX1 and MCRIP1[25]. Unlike previously identified FEM1B degrons, none of the C-degrons end with an Arg or contain a zinc-binding motif. Moreover, three out of five C-degrons (CCDC89, PSMB5, and BEX2) contained a proline at the C-terminus, and all contained an aromatic residue (Ψ) ~20 residues upstream

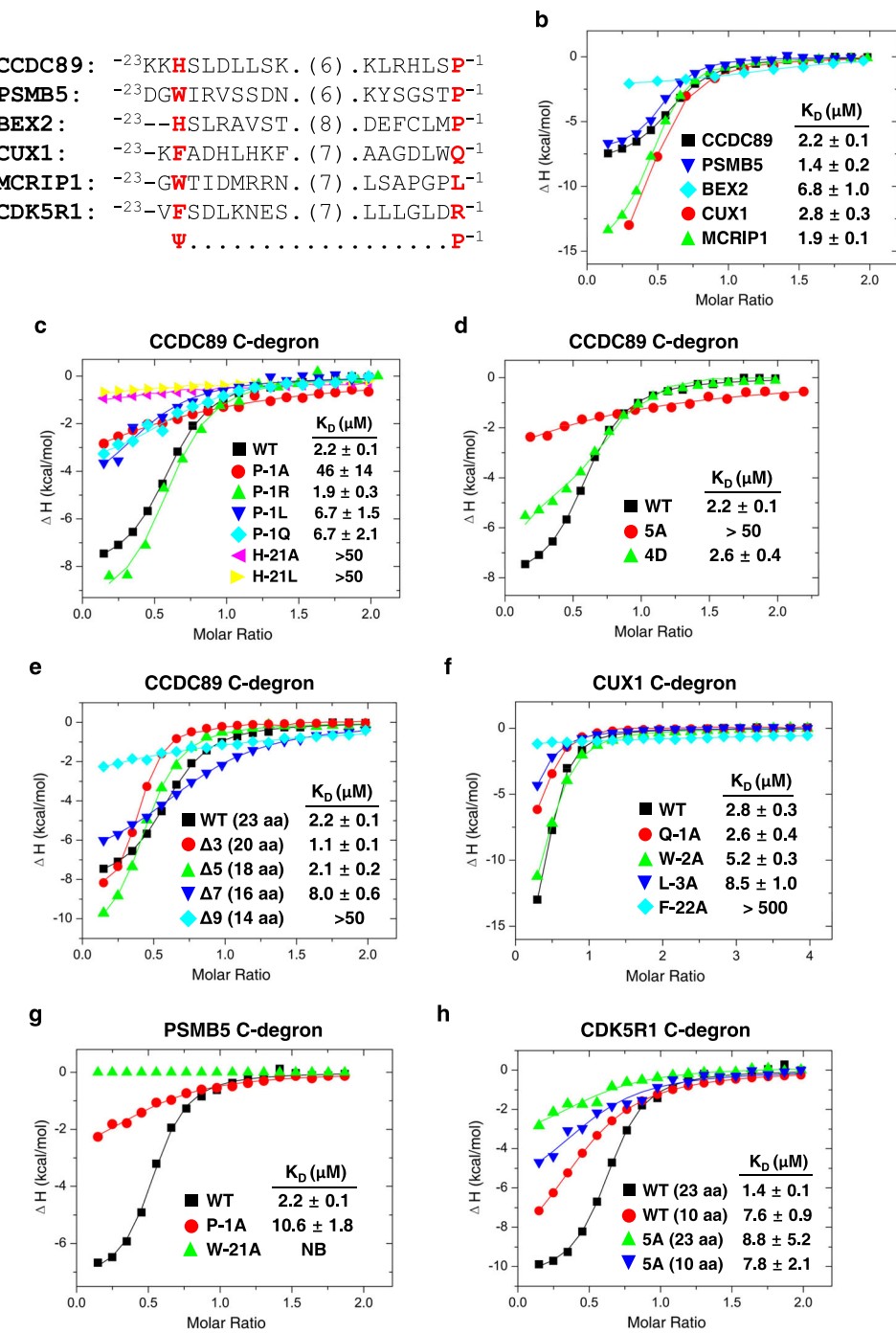

**Fig. 4 | Binding affinities of FEM1B-EB-EC and its variants towards various C-degrons. a** Sequence alignment of the C-degrons of CCDC89, CUX1, MCRIP1, PSMB5, BEX2 and CDK5R1, with Ψ indicating the specific aromatic residue or His. **b** ITC curves for the binding of the FEM1B-EB-EC ternary complex to different SUMO fusion C-degrons. **c** ITC curves for WT FEM1B-EB-EC binding to the CCDC89 C-degron and its variants. **d** ITC curves for CCDC89 C-degron binding to the FEM1B-EB-EC complex WT and the 5A or 4D mutants. **e** ITC curves for WT FEM1B-EB-EC binding to the CCDC89 C-degron and its deletion variants. **f** ITC curves for WT FEM1B-EB-EC binding to the CUX1 C-degron and its variants. **g** ITC curves for WT FEM1B-EB-EC binding to the PSMB5 C-degron and its variants. **h** ITC curves for the binding of 23-aa and 10-aa CDK5R1 C-degrons to FEM1B-EB-EC WT and the 5A mutant.

(Fig. 4a), suggesting that the C-degron recognition mode by FEM1B$^{FL}$ is different from our previously reported Arg/C-degron recognition mode by FEM1B$^{1-356}$, a shorter version of FEM1B that contains ANK1-7 and TPR1-3 (Fig. 1a).

To quantitatively examine the binding affinity between FEM1B and the above identified C-degrons, we expressed FEM1B with an EB/EC dimer to obtain a soluble ternary complex (FEM1B-EB-EC) (Supplementary Fig. 1f). The peaks of the purified WT and 4D mutant FEM1B-EB-EC complexes correspond to ~200 and 112 kDa, respectively (Supplementary Fig. 1g), suggesting that the majority of WT FEM1B-EB-EC behaves as a dimer in solution and that CUL2-RBX1 is not essential for CRL2$^{FEM1B}$ dimer formation. We then examined the degron binding affinities of FEM1B-EB-EC by iso-thermal titration calorimetry (ITC). The binding data indicate that the FEM1B-EB-EC complex binds to the five C-degrons with $K_D$s in the range of 1.4–6.8 μM (Fig. 4a, b).

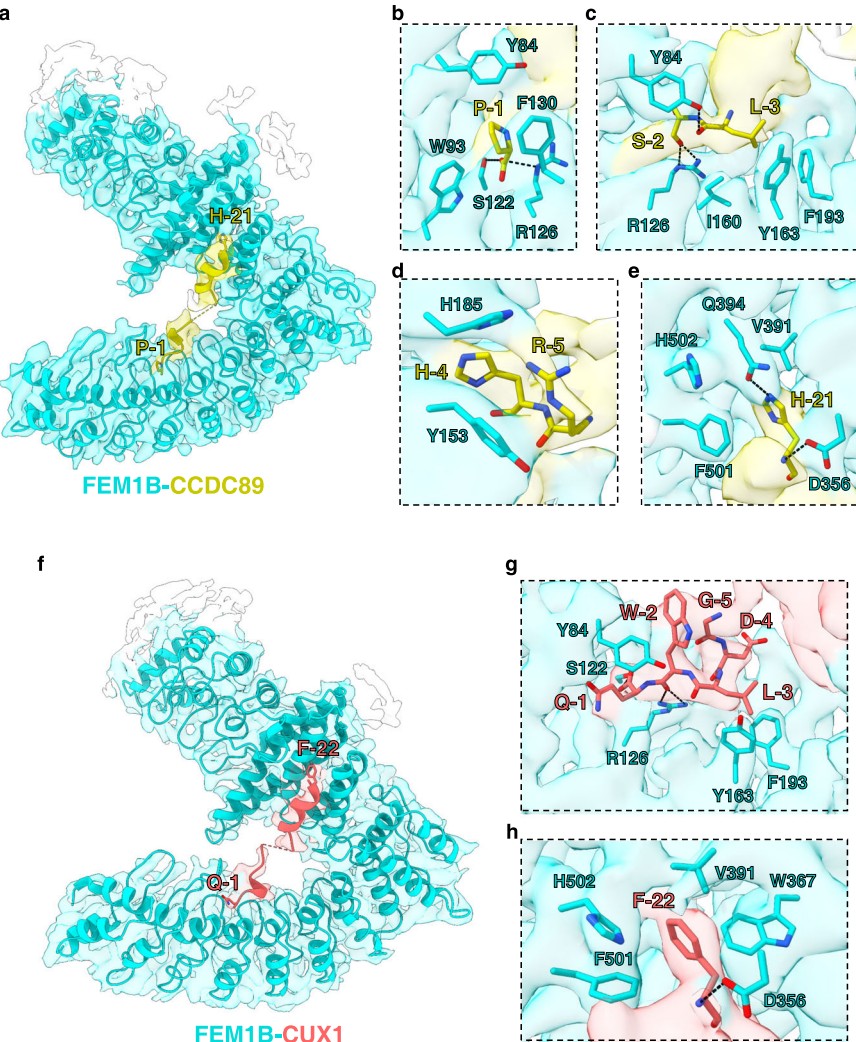

**Fig. 5 | Ψ-Pro/C-degron recognition by FEM1B. a** Overall structure and cryo-EM map of the FEM1B-CCDC89 complex. FEM1B and the CCDC89 peptide are shown in cyan and yellow cartoons, respectively, with the invisible peptide region indicated by dashes. Close-up views of FEM1B recognition of Pro-1 (**b**), Ser-2 and Leu-3 (**c**), His- 4 and Arg-5 (**d**), and His-21 (**e**). **f** Overall structure and cryo-EM map of the FEM1B- CUX1 complex. FEM1B and the CUX1 peptide are shown in cyan and salmon car- toons, respectively, with the invisible peptide region indicated by dashes. Close-up views of FEM1B recognition of Gln-1, Trp-2, Leu-3, Asp-4, Gly-5 (**g**), and Phe-22 (**h**).

## Recognition of CCDC89 C-degron by FEM1B

Due to the limited resolution of C-degron peptides observed in all solved cryo-EM structures, we performed local refinement for the [N8]CRL2[FEM1B] dimer[S1] to obtain a 3.55 Å resolution structure of the FEM1B- CCDC89 subcomplex (Supplementary Fig. 6 and Supplementary Table 1). Notably, the FEM1B-CCDC89 interface is conserved in all the CRL2[FEM1B]-CCDC89 complexes.

For convenience, the peptide is numbered from the C-terminus, with the last Pro being Pro-1, the penultimate Ser being Ser-2, etc. Most residues of CCDC89 are visible in the structure except for the five central residues ([−12]RELNG[−8]), probably due to their intrinsic flexibility (Supplementary Fig. 11a). The C-terminal region of the peptide adopts a $3_{10}$ helix structure and occupies the CDK5R1 Arg/C-degron binding site of FEM1B, which is formed by ANK3-4 (Fig. 5a and Supplementary Fig. 3). Pro-1 is accommodated by the same FEM1B pocket that has been reported to recognize the arginine at C-end. The main chain carbonyl group of Pro-1 is hydrogen bonded to the side chains of Ser122 and Arg126 of FEM1B, and its side chain forms hydrophobic interactions with Tyr84, Trp93, and Phe130 (Fig. 5b). The main chain carbonyl groups of Ser-2 and Leu-3 are hydrogen bonded to the side chains of Arg126 and Tyr84, respectively, with Leu-3 forming hydro- phobic interactions with Ile160, Tyr163, and Phe193 (Fig. 5c). His-4

forms hydrophobic interactions with His185, and Arg-5 interacts with Tyr153 via cation-π interactions (Fig. 5d). In contrast to [−5]RHLSP[−1], Leu- 6 and Lys-7 make few contacts with FEM1B residues.

The N-terminal portion of the CCDC89 peptide, [−22]KHSLDLLSKE[−13], adopts an α-helical conformation and interacts with TPR3, TPR4, and a $3_{10}$ helix insertion in ANK8 (Supplementary Fig. 3). Specifically, His-21 is positioned in a hydrophobic pocket composed of Val391, Phe501, and His502. Additionally, the imidazole ring and main chain nitrogen group of His-21 form hydrogen bonds with the side chains of Gln394 and Asp356, respectively (Fig. 5e).

Next, we employed mutagenesis and ITC binding assays to eval- uate the roles of CCDC89 C-degron residues in binding to FEM1B. The binding data demonstrated that while the P-1R mutant displayed binding affinity for FEM1B comparable to that of the WT peptide ($K_D$s: 1.9 μM vs. 2.2 μM), as expected, the P-1A, P-1Q, and P-1L substitutions decreased the FEM1B binding affinity by 3-20-fold ($K_D$s: 6.7−46 μM vs. 2.2 μM). Consistent with the suggested key role of CCDC89 H-21 in FEM1B binding, H-21A and H-21L reduced the binding affinity by >22- fold ($K_D$s: >50 μM vs. 2.2 μM) (Fig. 4c). Building upon our structural analysis that identified residues Asp356, Val391, Gln394, Phe501, and His502 of FEM1B as potential interaction points with His-21 of the CCDC89 C-degron (Fig. 5e), we further investigated the functional

relevance of these residues. To this end, we purified the FEM1B-EB-EC complex containing a FEM1B quintuple mutant, D356A/V391A/Q394A/F501A/H502A (5A mutant), and tested its binding to the CCDC89 C-degron. The binding data indicate that the affinity of the 5A mutant to CCDC89 is >22-fold weaker than that to WT FEM1B (K_Ds: >50 μM vs. 2.2 μM) (Fig. 4d). We further investigated the potential influence of CRL2$^{FEM1B}$ dimerization on substrate recognition. ITC binding data revealed that the CCDC89 peptide exhibits comparable binding affinity to both the WT FEM1B protein and the 4D mutant (K_Ds: 2.6 vs. 2.2 μM) (Fig. 4d). Our findings collectively demonstrate that Pro and Arg are favored at the extreme C-terminus, and both Pro-1 and His-21 within the CCDC89 C-degron motif are critical residues for its interaction with FEM1B. Furthermore, our data indicate that the dimerization state of CRL2$^{FEM1B}$ does not influence substrate recognition.

Our structural analyses suggest that the length of the C-degron might be critical for CCDC89 to engage the Ψ pocket and the Pro-1 binding pocket simultaneously, which inspired us to define the minimum distance between the two pockets. CCDC89-C-degron variants were generated by systematically deleting residues in the linker region ($^{-15}$SKERELNGK$^{-7}$), and their FEM1B binding affinities were examined (Supplementary Table 2). Although the deletion of up to five residues in the C-degron linker region does not disturb binding to FEM1B, the deletion of seven residues decreases the binding affinity by ~3.5-fold (K_Ds: 2.2 vs. 8.0 μM), and further deletion of two residues reduces the binding affinity by >20-fold (K_Ds: 2.2 vs. >50 μM) (Fig. 4e). This demonstrates that for the CCDC89 C-degron, the N-terminal histidine residue can be positioned between -15 and 20 residues upstream of the C-terminus without compromising its binding affinity to FEM1B.

### Recognition of the CUX1 C-degron by FEM1B

Although most of the identified FEM1B C-degrons bearing substrates contain a proline at the C-terminus[25], some of them, including CUX1, do not end with a proline. To understand how the C-degron of CUX1 is recognized by FEM1B, we further solved the cryo-EM structure of $^{un}$CRL2$^{FEM1B}$ with CUX1. The CRL2$^{FEM1B}$-CUX1 dimer$^a$ and dimer$^s$ were solved at resolutions of 3.38 Å and 3.27 Å, respectively (Supplementary Fig. 12 and Supplementary Table 1). The two dimeric structures of the $^{un}$CRL2$^{FEM1B}$-CUX1 complex are similar to those of the CRL2$^{FEM1B}$-CCDC89 complex (Supplementary Fig. 13). Local refinement was performed on the dimer$^a$ structure of $^{un}$CRL2$^{FEM1B}$-CUX1 to obtain a 3.60 Å resolution structure of the FEM1B-CUX1 subcomplex (Supplementary Fig. 12 and Supplementary Table 1).

The overall architecture of the FEM1B-CUX1 complex exhibits a high degree of structural similarity to that of the FEM1B-CCDC89 complex (Fig. 5f). Two parts of the CUX1 peptide, $^{-23}$KFADHLHKFH$^{-14}$ and $^{-7}$AAGDLWQ$^{-1}$, are visible, while the middle 6 residues, $^{-13}$ENDNGA$^{-8}$, are invisible (Supplementary Fig. 11c, d). The conformation of the CUX1 C-terminal fragment is slightly different from that of its counterpart in CCDC89. In contrast to the CCDC89 Pro-1 residue, the main chain carbonyl group of CUX1 Gln-1 is hydrogen bonded to the side chain of Ser122 in FEM1B. However, Gln-1 lacks residue-specific interactions with FEM1B itself. Trp-2 forms two hydrogen bonds with the side chain of FEM1B Arg126 via its main chain carbonyl group, with its side chain making π-π interactions with the aromatic ring of Tyr84; Leu-3 makes hydrophobic interactions with Tyr163 and Phe193; Asp-4 and Gly-5 make very few interactions with FEM1B residues except that the main chain carbonyl group of Asp-4 is hydrogen bonded to Arg126; and Gly-5 makes hydrophobic interactions with Trp-2 to stabilize the CUX1 conformation (Fig. 5g).

The CUX1 N-terminal fragment adopts a helix similar to its counterpart in CCDC89. Phe-22 is snugly accommodated into the pocket that was identified earlier as the binding site for CCDC89 His-21. Phe-22 forms hydrophobic interactions with FEM1B Val391 and His502, π-π interactions with Phe501 and Trp367, and a main chain hydrogen bond with Asp356 (Fig. 5h). The favorable π-π interactions suggest that the

pocket likely prefers aromatic residues, such as Phe, Tyr, and Trp, over other residues. However, as observed for CCDC89, a His could interact with the pocket residues with its imidazole ring and form a hydrogen bond with Gln394 (Fig. 5e), positioning it as another favorable residue for recognition by FEM1B.

To evaluate the roles of CUX1 C-degron residues in FEM1B binding, we generated several CUX1 mutants and examined their FEM1B binding affinities by ITC. While the Q-1A mutant binds FEM1B with a similar affinity to WT CUX1, the W-2A and L-3A mutations reduced the binding affinity by ~1.8-fold (K_Ds: 5.2 vs. 2.8 μM) and ~3.0-fold (K_Ds: 8.5 vs. 2.8 μM), respectively (Fig. 4f). F-22A decreased the binding affinity by > 170-fold (K_Ds: >500 vs. 2.8 μM) (Fig. 4f), suggesting that Phe-22 plays a critical role in binding to FEM1B. To understand whether the key roles of the C-terminal proline and upstream aromatic residue also apply to other identified C-degrons, we chose to examine the binding affinities of the variants of the PSMB5 C-degron because it contains both Pro-1 and Trp-21. In agreement with previous analysis, while P-1A reduced FEM1B binding by ~4.8-fold, W-21A abolished the binding (Fig. 4g).

Taken together, structural analysis of the C-degron-bound FEM1B complexes indicates that an upstream aromatic residue (Phe, Tyr, Trp) or a histidine is critical for binding to FEM1B. The C-terminal proline preference observed in most FEM1B C-degrons and its established role in substrate instability[25] suggest that the C-terminal proline is favored. However, it is less critical for binding, as some substrates, such as CUX1, do not end with a proline. Notably, when Pro-1 is absent, a penultimate hydrophobic residue, such as Trp-2, might compensate, as observed in the CUX1 C-degron. Therefore, we named the C-degron characterized here as Ψ-Pro/C-degron with Ψ indicating aromatic residues and His.

Our previous study reported that the Arg/C-degron of CDK5R1, which contains the last 10 residues, is sufficient to interact with FEM1B[1-3,5,6,20]. Intriguingly, sequence alignment with the last 23 amino acids of CDK5R1 demonstrated that CDK5R1 also contains the upstream aromatic residue Phe (Phe-22) (Fig. 4a). We then employed an ITC binding assay to examine the FEM1B binding affinities of the CDK5R1 C-degrons of varying lengths. The binding data indicate that the longer fragment of the CDK5R1 C-degron (23 aa) binds to FEM1B > 5-fold more strongly than the 10-aa fragment (K_Ds: 1.4 vs. 7.6 μM). In contrast, both C-degrons bind to the FEM1B Ψ pocket mutant (5A) with K_Ds in the range of 7.8–8.8 μM (Fig. 4h). This finding suggests that Phe-22 could increase the binding affinity of CDK5R1 for FEM1B by contacting the Ψ pocket. A full recognition sequence of Ψ-Arg/C-degron is likely necessary for CDK5R1 to achieve optimal FEM1B binding affinity.

### Validation of the FEM1B mode of substrate recognition utilizing in vitro and in vivo assays

To study how the FEM1B-C-degron interaction and FEM1B-mediated dimerization and neddylation affect the E3 activity of CRL2$^{FEM1B}$, we employed an in vitro ubiquitination assay to examine the PolyUb of CCDC89 by CRL2$^{FEM1B}$ and its variants. First, we compared the activity of $^{un}$CRL2$^{FEM1B}$ and its neddylated form. The assay showed that the activity of $^{un}$CRL2$^{FEM1B}$ was ~60% that of its neddylated counterpart (Supplementary Fig. 14a, b). This finding implies that CUL2 neddylation serves as an activation mechanism for E3 activity. Next, we introduced three single point mutations in CCDC89, namely, P-1A, L-3E, and H-21A. These mutations resulted in a 50–70% reduction in PolyUb by the E3 complex (Supplementary Fig. 14c–f), underscoring the important roles of these C-degron residues in FEM1B binding. Consistent with the ITC data, mutations of the FEM1B residues involved in Pro-1 or His-21 binding also diminished the activity of CRL2$^{FEM1B}$ E3 (Supplementary Fig. 14e–h). Notably, the FEM1B dimerization mutant (4D), which does not affect CCDC89 binding, slightly weakened E3-mediated PolyUb (Supplementary Fig. 14i, j), suggesting that

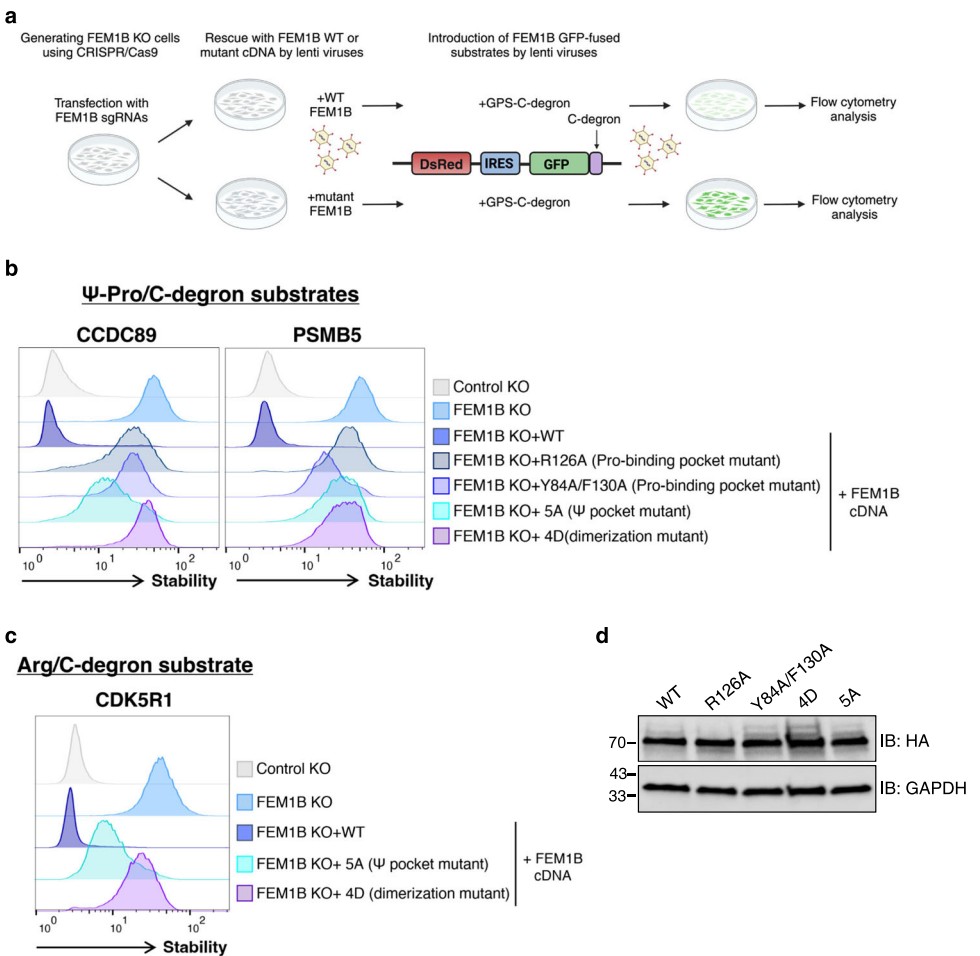

**Fig. 6 | GPS assay to validate FEM1B residues critical for degron binding and dimerization of the E3 complex. a** Overview of the GPS experiment used to investigate the effect of WT and mutant FEM1B proteins on the stability of GFP-fused substrates bearing C-degrons. The figure was created with BioRender.com. Stability analysis of the indicated GFP-fused 23-mers bearing a Ψ-Pro/C-degron (**b**) or Arg/C-degrons (**c**) in various genetic backgrounds. The GFP/DsRed ratio was used to indicate the stability of the GFP-fused C-degron and was analyzed by flow cytometry. In each case, GFP-fused degron instability was rescued in cells expressing the WT but not the mutant FEM1B proteins. Control KO cells (cells transfected with sgRNA targeting AAVSI) expressing GPS substrate served as the reference for substrate instability. **d** The abundance of exogenously expressed FEM1B protein, WT or the indicated mutants, as assessed by immunoblotting (IB) using an anti-HA antibody. GAPDH was used as a loading control. Source data are provided as a Source Data file.

a potential role for CRL2$^{FEM1B}$ dimerization in facilitating Ub transfer rather than in substrate recognition. Altogether, the results of the ubiquitination assay indicated that both the C-terminal and upstream aromatic/His residues act as the key C-degron residues interacting with FEM1B, and the assembly of the CRL2$^{FEM1B}$ homodimer is required to achieve maximal E3 activity toward C-degron-bearing substrates.

To validate the functional role of the C-degron-binding residues in FEM1B in vivo, we constructed a series of FEM1B mutants in a lentiviral expression vector and tested their activity in a cell-based system termed GPS profiling[32]. GPS is a fluorescence-based reporter system that measures protein stability in live cells. This system is based on a bicistronic lentiviral expression vector that encodes DsRed, which serves as an internal expression reference for the lentiviral cassette, and a GFP fusion that is translated from an internal ribosome entry site (IRES). In this method, the GFP/DsRed ratio serves to determine the effect of the fusion partner on the stability of GFP (Supplementary Fig. 15a)[17,32].

For cell-based assays, we first generated HEK293T FEM1B knock-out (KO) cells using CRISPR/Cas9. We then transduced the KO cells with lentiviral constructs encoding WT or mutant FEM1B cDNA, followed by recovery and a second transduction with GPS peptide substrates (Fig. 6a). As examples of FEM1B substrates, we constructed GPS

reporters encoding the 23 C-terminal residues ("23-mers") of PSMB5 and CCDC89 harboring Ψ-Pro/C-degrons. Additionally, a control reporter containing the C-terminus of CDK5R1, which features an Arg/C-degron motif, was also constructed.

Consistent with the structural analysis and in vitro ubiquitination assays, mutations disrupting the Pro-1 binding pocket (R126A and Y84A/F130A) and the aromatic binding pocket failed to restore substrate degradation, as shown in Fig. 6b. Likewise, the aromatic binding pocket mutant displayed impaired turnover of the Arg/C-degron substrate CDK5R1 (Fig. 6c), corroborating the ITC data (Fig. 4h). Notably, immunoblot analysis confirmed comparable expression levels of the mutants and WT FEM1B (Fig. 6d). The activity of FEM1B toward Ψ-Pro/C-degrons was dependent on the two substrate binding pockets; P-1A or H-21A substitutions resulted in substrate stabilization to a similar extent as that observed in FEM1B KO cells, and FEM1B cDNA expression failed to promote substrate degradation (Supplementary Fig. 15b). Altogether, the GPS cell-based assay demonstrated the importance of the FEM1B C-degron binding pockets in Ψ-Pro/C-degron recognition in vivo.

Finally, we employed the FEM1B 4D mutant, the dimerization mutant, to assess the functional significance of CRL2$^{FEM1B}$ dimerization for C-degron binding and substrate turnover. While exhibiting slightly

reduced E3 activity in vitro (Supplementary Fig. 14i, j), the 4D mutant displayed impaired in vivo degradation of GFP-fusion substrates harboring either Ψ-Pro/C-degrons (Fig. 6b) or the Arg/C-degron (Fig. 6c). These data suggest that CRL2[FEM1B] dimerization is essential for substrate PolyUb and subsequent turnover of substrates bearing Ψ-Pro/C-degron or Arg/C-degrons.

## Discussion

CRL2[FEM1B] was previously shown to recognize distinct Arg/C-degrons and Cys-dependent degrons. Two recent studies revealed that FEM1B additionally targets an additional class of C-terminal proline-ending degrons and some degrons containing the internal WxxYL motif [25,26]. Here, we present cryo-EM structures of CRL2[FEM1B] in complex with various degron peptides, including those derived from both unmodified and neddylated CRL2[FEM1B], to elucidate the mechanisms underlying FEM1B's degron specificity and CRL2[FEM1B] complex activation.

Unexpectedly, unmodified CRL2[FEM1B] is a homodimer exhibiting symmetric and asymmetric dimerization states, dimer[a] and dimer[s], respectively. Neddylated CUL2 retains the ability to form dimers in two distinct conformations. While the overall architecture of the dimers is preserved between the unmodified state (dimer[s]) and the neddylated symmetric state (dimer[S1]), the WHB domains are not visible in the latter structure. The conjugation of NEDD8 disrupts the dimer[a] assembly by introducing potential steric clashes, causing structural rearrangement to form a different dimeric assembly, dimer[S2] (Supplementary Fig. 16). At the molecular level, the binding of E2-Ub leads to potential steric clashes in all solved structures, suggesting that the conformation of RBX1 must change, including its dissociation from the current ligand, to interact with E2-Ub. NEDD8 facilitates RBX1 accessibility to E2-Ub by displacing the WHB domain (Supplementary Fig. 10). Thus, the likely role of CUL2 neddylation is to lower the energy barrier for the interaction of E2-Ub with RBX1.

Despite the different dimeric states in [un]CRL2[FEM1B] and [NS]CRL2[FEM1B], there are several common shared structural features in all solved structures. First, the intact assembly of CRL2[FEM1B] protomers is maintained in all dimeric structures. For each protomer, FEM1B interacts with EC and CUL2 via major and minor sites through hydrophobic interactions. Second, a hydrophobic patch located towards the C-terminus of FEM1B is required for the dimerization of CRL2[FEM1B] in all states. Third, the C-degrons of CCDC89 and CUX1 interact with both the N- and C-arms of FEM1B, suggesting that both the Ψ and Pro-binding pockets are critical for docking this class of C-degrons to FEM1B.

To compare the various FEM1B-degron binding modes, we superimposed the structures of FEM1B-CCDC89, FEM1B-CUX1, FEM1B-CDK5R1 (PDB: 7CNG), and FEM1B-FNIP1 (PDB: 7ROY) (Supplementary Fig. 17a, b). All ligands, except FNIP1, represent C-degrons, with most of their C-terminal residues binding to the same pocket of FEM1B. Their binding preference follows the order Arg-1 > Pro-1 > Gln-1 (Supplementary Fig. 17c–e). While Arg-1 forms both cation-π and hydrogen bonding interactions with the pocket residues and Pro-1 makes extensive hydrophobic interactions with the pocket residues (Supplementary Fig. 17c, d), Gln-1 does not form any residue-specific interactions with FEM1B (Supplementary Fig. 17e). In contrast, no FNIP1 residues are engaged in the pocket accommodating C-terminal residues; instead, they form intermolecular zinc fingers with FEM1B (Supplementary Fig. 17f). FEM1B recognizes the Arg/C-degron of CDK5R1 similarly to CCDC89 and CUX1. In all three cases, an aromatic residue (Phe-22 in CDK5R1 and CUX1, His-21 in CCDC89) provides additional interactions with the Ψ pocket of FEM1B (Supplementary Fig. 17g, h). Notably, for CCDC89 and CUX1, these interactions likely compensate for the weaker binding affinity caused by the lack of a C-terminal arginine. The flexibility of FEM1B in substrate recognition might be required for regulating the turnover of diverse substrates in distinct pathways.

Another intriguing feature of CRL2[FEM1B] is that although dimerization does not affect its binding to these identified C-degrons, dimerization does have an impact on the degradation of substrates. Two potential mechanisms for CRL2[FEM1B]-mediated protein ubiquitination have been proposed. In trans-acting mode, C-degron binding by one CRL2[FEM1B] protomer could facilitate ubiquitin transfer catalyzed by RBX1 of the other protomer. Alternatively, the two CRL2[FEM1B] protomers might cooperate to catalyze the ubiquitination of an oligomeric substrate. Notably, two potential FEM1B substrates, CCDC89 and CDK5R1, are known to form dimers or higher-order oligomers [33,34]. In addition, in vivo, monomeric CRL2[FEM1B] might exhibit enhanced susceptibility to disruption by negative regulators, including deubiquitinases (DUBs) and deneddylation-associated factors, such as CSN complexes and CAND1. Further study is required to understand whether CRL2[FEM1B] dimerization also impacts its E3 activity towards other substrates in vivo. Overall, the present study reveals the assembly modes of CRL2[FEM1B], as well as its ability to recognize distinct C-degrons, with implications in PROTAC design for future TPD.

## Methods

### Cloning, protein expression and purification

All proteins are of human origin. All genes used in this study for protein expression and purification, except for UBA1, were amplified by PCR from a complementary DNA library. FEM1B, UBA1, CDC34, ubiquitin and UBE2M were cloned and inserted into modified pET28a vectors fused with the open reading frame of the yeast *SMT3* gene. EB-EC (residues 17–112) was cloned and inserted into the pRSFDuet-1 vector. APPBP1-UBA3 were cloned and inserted into the pETDuet-1 vector. NEDD8 and FLAG-CCDC89 were cloned and inserted into the His6-pGEX 4T-1 vector. The C-term 23-mer peptides CCDC89, CUX1, PSMB5, BEX2, CDK5R1 and MCRIP1 were cloned and inserted into the pET28a-SUMO vector via overlap extension PCR to obtain SUMO fusion peptides, which were fused via a (Gly-Gly-Gly-Ser)₂ linker. The CUL2Δ·RBX1 construct (CUL2Δ: Δ117-134) was a kind gift from Dr. Xing Liu at Purdue University.

All recombinant proteins were overexpressed in *Escherichia coli* BL21(DE3) cells. FEM1B and EB-EC were co-transformed with two vectors. The cells were grown in LB media at 37 °C until the optical density (OD₆₀₀) reached approximately 0.8. Protein expression was induced with 0.2 mM β-d-1-thiogalactopyranoside for 20 h at 16 °C. The cells were collected by centrifugation at 3600 × g for 10 min at 4 °C, after which the pellets were resuspended in lysis buffer containing 20 mM Tris, 400 mM NaCl, and 2 mM imidazole (pH 7.5). Recombinant proteins were purified by Ni-NTA (GE Healthcare), and eluted with 20 mM Tris, 400 mM NaCl, 500 mM imidazole, pH 7.5.

Fractions containing target proteins were incubated overnight at 4 °C with TEV protease to cleave the tag and further purified by size-exclusion chromatography in 25 mM HEPES, 150 mM NaCl, pH 7.5, and 1 mM DTT. The CUL2Δ·RBX1 complex was neddylated by mixing 0.2 μM NAE1 (APPBP1-UBA3), 1 μM UBE2M, 12 μM CUL2-RBX1, and 25 μM NEDD8 in 25 mM HEPES (pH 7.5), 150 mM NaCl, 10 mM MgCl₂, and 1 mM ATP. The reaction was performed at 37 °C. Neddylation was quenched after 10 min by the addition of 10 mM DTT. Neddylated CUL2-RBX1 was purified by size-exclusion chromatography in 25 mM HEPES, 150 mM NaCl, pH 7.5, and 1 mM DTT. Fractions containing Sumo fusion peptides were incubated overnight at 4 °C with ULP1 protease to cleave the tag and further purified by anion exchange chromatography using a HiTrap Q HP column or a HiTrap S HP column (Cytiva Life Sciences). Proteins for ITC were purified by size-exclusion chromatography in 20 mM Tris, 200 mM NaCl, pH 7.5, and 1 mM EDTA. All variants of FEM1B, CCDC89, PSMB5 and CUX1 were generated using PCR and verified by sequencing. All variants were purified in the same way as the WT protein.

## Isothermal titration calorimetry (ITC)

ITC experiments were performed on a MicroCal iTC200 calorimeter (GE Healthcare) at 25 °C by titrating 2 μl of peptides (0.5–1 mM) into cells containing 50 μM proteins, with a spacing time of 120 s and a reference power of 5 μCal s$^{-1}$. Control experiments were performed by titrating peptides (0.5–1 mM) into buffer, and the concentrations were subtracted during analysis. Binding isotherms were plotted, analyzed and fitted based on a one-site binding model by MicroCal PEAQ-ITC analysis software (Malvern Panalytical). The SUMO fusion peptides used for ITC experiments are summarized in Supplementary Table 2. The original ITC binding data are shown in Supplementary Fig. 18.

## In vitro ubiquitylation assays

In vitro ubiquitination assays were performed in a 40 μl volume with 0.15 μM UBA1, 1 μM CDC34, 1 μM neddylated CUL2-RBX1 or 1 μM CUL2-RBX1, 1 μM FEM1B-ELOB-ELOC, 1 μM Flag-CCDC89 and 30 μM Ub at 37 °C in reaction buffer (25 mM HEPES, pH 7.5, 100 mM NaCl, 10 mM MgCl$_2$, 1 mM DTT and 5 mM ATP). The reactions were terminated by adding 5× sodium dodecyl sulfate (SDS) sample buffer and then resolved by SDS–PAGE. Ubiquitinated products were detected by immunoblotting. The same concentrations of FEM1B and Flag-CCDC89 variants were used in the assay as the respective wild type. The following antibodies were used: anti-FLAG DYKDDDDK tag (Cell Signaling Technology, Cat#2368, 1:1,000 dilution) and HRP-linked anti-rabbit IgG (Cell Signaling Technology, Cat#7074, 1:1,000 dilution). The experiments were performed three times with similar results, and the quantification was performed using ImageJ, with the data subsequently analyzed with GraphPad Prism 8.

## Multiangle static light scattering (MALS)

The molecular mass analysis of the $^{N8}$CRL2$^{FEM1B}$ WT and the variants was performed on an AKTA Pure system (GE Healthcare) coupled with a DAWN HELEOS 8+ instrument (Wyatt Technology). One hundred microliters of protein sample (5 mg/ml) was loaded into a Superose 6 Increase 10/300 GL column (GE Healthcare) pre-equilibrated in 25 mM HEPES, pH 7.5, 150 mM NaCl, and 1 mM DTT. The data were analyzed with ASTRA software (Wyatt).

## Cryo-EM sample preparation and data collection

Cryo-EM samples were generated by mixing 10 μM neddylated CUL2-RBX1 or 10 μM CUL2-RBX1, 12 μM FEM1B-ELOB-ELOC, and 50 μM C-term 23-mer peptide (CCDC89, CUX1 or CDK5R1). The mixture was incubated at 4 °C for at least 30 min and purified using a Superose 6 Increase 10/300 GL column (GE Healthcare) in 25 mM HEPES (pH 7.5), 150 mM NaCl, and 1 mM DTT. The peak fractions were concentrated to 5 mg ml$^{-1}$. Next, 4 μl of freshly assembled protein complex was supplemented with 0.2 mM fluorinated octyl maltoside (Anatrace) and immediately applied onto a glow-discharged (20–30 s) 200-mesh R2/1 Quantifoil copper grid, incubated for 20 s, blotted for 3–3.5 s and plunge frozen in liquid-nitrogen-cooled liquid ethane using a Vitrobot Mark IV (Thermo Fisher Scientific) operated at 4 °C and 100% humidity.

The electron micrographs for all datasets were acquired on a 300 kV FEI Titan Krios electron microscope equipped with a K3 Summit direct electron camera (Gatan). Automated data acquisition for all datasets was performed using EPU software (Thermo Fisher) at a magnification of 105,000× (corresponding to a calibrated sampling of 0.82 Å per pixel). A total of 4,472 micrographs of the CRL2$^{FEM1B}$-CCDC89 complex, 4,223 micrographs of the $^{N8}$CRL2$^{FEM1B}$-CCDC89 complex, 4,232 micrographs of the $^{N8}$CRL2$^{FEM1B}$-CDK5R1 complex, and 8,112 micrographs of the $^{un}$CRL2$^{FEM1B}$-CUX1 complex were collected. The cumulative dose used was 57.6 e$^-$ Å$^{-2}$, which was fractionated into 30 frames. Target defocus values typically ranged from −1.5 to −2.9 μm.

## Image processing

All image processing and analyses were performed with cryoSPARC. Movies were aligned using patch motion correction followed by contrast transfer function (CTF) estimation in cryoSPARC. The particles were picked manually via automatic picking. All classifications and refinements were conducted in cryoSPARC. For the unmodified CRL2$^{FEM1B}$-CCDC89 complex, a total of 1,577,333 particles were autopicked and extracted in cryoSPARC, and then the extracted particles were subjected to 2D classification, with good classes selected for 3D classification. A total of 1,171,468 particles were used for ab initio 3D reconstruction in cryoSPARC into four classes. Classes exhibiting CUL2 and FEM1B density were subjected to heterogeneous refinement in cryoSPARC. Then, nonuniform refinement together with local and global CTF refinement was performed with the two good classes, yielding two maps with 3.37 Å resolution from 470,742 particles and 3.39 Å resolution from 236,621 particles.

For the $^{N8}$CRL2$^{FEM1B}$-CCDC89 complex, the image processing steps are similar to those described above, finally yielding maps with 4.09 Å resolution from 102,432 particles and 3.44 Å resolution from 84,450 particles. The final map of the FEM1B-CCDC89 complex was obtained by local refinement with a resolution of 3.55 Å.

For the $^{N8}$CRL2$^{FEM1B}$-CDK5R1 complex, the map of the $^{N8}$CRL2$^{FEM1B}$-CCDC89 complex was used as a template for particle picking and heterogeneous refinement in cryoSPARC, yielding a map with 3.54 Å resolution from 447,970 particles.

For the $^{un}$CRL2$^{FEM1B}$-CUX1 complex, the 3.27 Å resolution map of the dimer$^a$ and the 3.38 Å resolution map of the dimer$^s$, were obtained from 420,437 and 206,765 particles, respectively. The structure of FEM1B-CUX1 was achieved by local refinement with a resolution of 3.51 Å. Map resolution was estimated by the "gold standard" Fourier shell correlation (FSC) at the 0.143 criterion. Local resolutions were estimated using the Local Map Estimation program in cryoSPARC, with the local resolution map depicted by UCSF Chimera[35].

## Model building and refinement

The sharpened maps were generated by cryoSPARC. The atomic models for model building, including those of FEM1B, EB, EC, CUL2, and RBX1, were obtained from AlphaFold2[36]. The degron peptides were fitted into the map after the other CRL2$^{FEM1B}$ subunits were built by using COOT 0.8.9.2[37] and were further refined in Phenix 1.19.2 by using real-space refinement[38]. The refined structures were evaluated with MolProbity in Phenix 1.19.2[38]. The refinement statistics are summarized in Supplementary Table 1. PyMOL 1.7.0.1 (https://pymol.org/) and UCSF Chimera 1.11.2[35] were used for figure preparation.

## Mammalian expression plasmids

To express FEM1B in mammalian cells, cDNA for FEM1B was obtained from the Ultimate ORF Clone collection (Thermo Fisher Scientific) and subcloned and inserted into the lentiviral pHAGE-Flag-HA Gateway Destination vector or MYCx5 Gateway Destination vector via an LR recombination reaction (Thermo Fisher Scientific, #11791020). Mutations of key residues in the degron binding pocket or the dimerization interface of FEM1B were generated by PCR-mediated site-directed mutagenesis using the QuikChange Lightning Kit (Agilent, #210518) according to the manufacturer's protocol. All mutations were confirmed by Sanger sequencing. The primer sets used to generate FEM1B mutants for expression in mammalian cells are listed in Supplementary Table 3.

For individual CRISPR/Cas9-mediated gene disruption experiments, the lentiCRISPR v2 vector was used (Addgene #52961). Oligonucleotides encoding the top and bottom strands of the sgRNAs were synthesized (IDT), annealed, cloned and inserted into the lentiCRISPR v2 vector as previously described[17]. The nucleotide sequences of the sgRNAs used were as follows:

sg-AAVSI: GGGGCCACTAGGGACAGGAT
sg-FEM1B: GGCTGCACACCAAAGAGCAG

## Cell lines

HEK293T (ATCC CRL-3216) cells were maintained at 37 °C and 5% $CO_2$ in Dulbecco's modified Eagle's medium (DMEM) (Life Technologies) supplemented with 10% fetal bovine serum (HyClone), 100 units/ml penicillin and 0.1 mg/ml streptomycin (Thermo Fisher Scientific).

## Transfection and lentivirus production

Lentiviral stocks were generated through the transfection of HEK293T cells with the lentiviral transfer vector plus plasmids encoding Gag-Pol, Rev, Tat and VSV-G using PolyJet in vitro DNA Transfection Reagent (SignaGen Laboratories) as recommended by the manufacturer. Lentiviral supernatants were collected 48 h later, passed through a 0.45 mm filter, and applied to target cells in the presence of 8 mg/ml hexadimethrine bromide (Polybrene).

## Generation of CRISPR/Cas9 knockout cells

HEK293T cells were transfected with sgRNAs targeting AAVSI or FEM1B as previously described[17] to generate control or FEM1B KO cells, respectively. Forty-eight hours after transfection, the cells were selected with puromycin for 48 h to eliminate nontransfected cells. KO efficiency was analyzed 7 days posttransfection using functional GPS assays with known FEM1B substrates as described previously[17].

## Global protein stability (GPS) assay

The GPS assay using GFP-fused C-degron reporter cell lines was described previously[17]. Briefly, the last 23 amino acids of CCDC89 (KKHSLDLLSKERELNGKLRHLSP*) and PSMB5 (DGWIRVSSDNVADLHE-KYSGSTP*) were amplified by PCR from the C23mer peptidome library[17] and cloned and inserted into the GPS vector downstream of GFP using the BstBI and XhoI sites. To test the activity of FEM1B mutants using the GPS assay, lentiviral vectors encoding WT or mutant versions of FEM1B were packaged and introduced into FEM1B KO HEK293T cells, followed by selection with puromycin (1 μg/ml) for 3 days to generate rescue cells stably expressing the FEM1B proteins. Stable FEM1B cells were then subjected to a second round of transduction with the GPS FEM1B sub-strate reporters. The GPS reporters were packaged and used to trans-duce stable rescue cells, followed by blasticidin (20 μg/ml) selection for 3 days. Flow cytometry was used to record the fluorescence signals of GFP and DsRed from cells transduced with the GPS reporters. Analysis of the GFP/DsRed ratio was performed using FlowJo.

## Western blotting analyses

Cells were lysed in ice-cold lysis buffer (10 mM $Na_2HPO_4$, 100 mM NaCl, 5 mM EDTA pH 8, 1% Triton X-100, 0.5% deoxycholic acid sodium salt, 0.1% SDS) supplemented with Halt™ Protease and Phosphatase Inhibitor Cocktail (Thermo Scientific) for 25 min at 4 °C. Lysates were clarified by centrifugation (20,000 × g, 15 min, 4 °C), and nuclear pellets were resuspended in lysis buffer, sonicated briefly, and reclarified. The protein concentration was determined by a standard Bradford assay (Bio-Rad #500-0006), a linear bovine serum albumin (BSA) calibration curve, and an Epoch microplate spectrophotometer. Proteins were subsequently resolved by SDS–PAGE (Mini-PROTEAN TGX Precast Protein Gels, Bio-Rad) and transferred to a nitrocellulose membrane (Trans-Blot Turbo System, Bio-Rad), which was then blocked in 10% nonfat dry milk in PBS + 0.1% Tween-20 (PBS-T). The membrane was incubated with primary antibodies, including rabbit anti-GAPDH (Cell Signaling Technology, Cat# 2118, 1:1000), rabbit anti-Myc-Tag (71D10) (Cell Signaling Technology, Cat#2278) , rabbit anti-HA (Cell Signaling Technology, Cat# 3724, 1:1,000), and rabbit anti-CUL2 (Bethyl Laboratories®, Cat# A302-476A, 1:1000) overnight at 4 °C, and then, following three washes with PBS-T, HRP-conjugated goat anti-rabbit IgG (H+L) secondary antibody (Jackson ImmunoResearch Labs, Cat# 111-035-144, 1:20,000) was added for 1 h at room temperature. Fol-lowing an additional three washes in PBS-T, reactive bands were visua-lized using SuperSignal West Femtochemiluminescence substrate (Pierce; #34095) or EZ-ECL (Biological Industries; #20-500-171) for 5 min. Reactive bands were visualized using ImageQuant TL software v8.2 on an Amersham Imager 680 (Cytiva).

## Immunoprecipitation

HEK293T FEM1B KO cells were engineered to stably express single-copy integrants of HA-tagged FEM1B, either wild-type (WT) or 4D mutant, using lentiviral transduction. For co-immunoprecipitation experiments with endogenous CUL2, 10 million cells in one 10 cm plate of WT or 4D mutant was used per immunoprecipitation experiment. For co-immunoprecipitation experiments with MYC-FEM1B, 4 million cells stably expressing HA-FEM1B (WT or 4D mutant) were seeded per 10 cm plate. After 24 h, the cells were transfected with 5 μg of MYC-FEM1B plasmid. Following 48 h of recovery, the cells were detached from the plates with trypsin and lysed in ice-cold lysis buffer (50 mM Tris pH 8, 150 mM NaCl, 1% NP40) supplemented with Halt™ Protease and Phosphatase Inhibitor Cocktail (Thermo Scientific, Cat# 78440) for 25 min at 4 °C. Lysates were clarified by centrifugation (20,000 × g, 15 min, 4 °C) followed by immunoprecipitation using HA- magnetic agarose beads (Pierce™ Anti-HA Magnetic Beads Cat #88836), which were added to the supernatants and incubated with rotation for 2 h at 4 °C. The beads were then washed three times with lysis buffer before bound proteins were eluted upon incubation with HA peptide (Sigma-Aldrich, Cat #I2149) for 30 min at 37 °C. Proteins were subsequently resolved by SDS-PAGE as previously described.

## Gel filtration chromatography of $CRL2^{FEM1B}$

Size-exclusion chromatography was performed on proteins extracted from HEK293T FEM1B KO cells stably expressing either WT FEM1B or the 4D mutant. Cells from one 15 cm plate were harvested with 700 μl of PBS using a cell scraper. Cell lysates were prepared by homogenization with a dounce homogenizer followed by ultracentrifugation at 50,000 rpm. The soluble material was loaded on preequilibrated Superose 6 10/300 column (Cytiva Lifescience™). Then, 500 μl fractions were collected for each sample and protein elution was monitored via UV absorbance of the eluate at 280 nm. 50 μl from each fraction was subjected to SDS–PAGE, followed by western blot using HA antibody. ImageJ was used to quantify the band intensity across all fractions, and the results are presented in Supplementary Fig. 9b.

## Flow cytometry

HEK293T cells were detached with trypsin, and washed once with PBS. Flow cytometry data were collected using an LSRFortessa cell analyzer (BD Biosciences) or CytoFlex S (Beckman Coulter), and were analyzed using FlowJo v.10 (BD Biosciences).

## Statistics and reproducibility

Each flow cytometry analysis and western blot presented here are representative examples of at least two independent experiments with consistent results, demonstrating reproducibility.

## Reporting summary

Further information on research design is available in the Nature Portfolio Reporting Summary linked to this article.

## Data availability

Cryo-EM maps were deposited in the Electron Microscopy Data Bank under accession codes EMD-37736 ($^{un}$CRL2$^{FEM1B}$-CCDC89 dimer$^s$), EMD-37737 ($^{un}$CRL2$^{FEM1B}$-CCDC89 dimer$^a$), EMD-37739 ($^{NS}$CRL2$^{FEM1B}$-CDK5R1 dimer$^{S2}$), EMD-37740 (FEM1B-CCDC89 complex), EMD-37742 ($^{un}$CRL2$^{FEM1B}$-CUX1 dimer$^s$), EMD-37743 ($^{un}$CRL2$^{FEM1B}$-CUX1 dimer$^a$), EMD-37744 ($^{NS}$CRL2$^{FEM1B}$-CCDC89 dimer$^{S2}$), EMD-37745 ($^{NS}$CRL2$^{FEM1B}$-CCDC89 dimer$^{SI}$), and EMD-37746 (FEM1B-CUX1 complex). Atomic coordinates have been deposited into the PDB under accession numbers 8WQA ($^{un}$CRL2$^{FEM1B}$-CCDC89 dimer$^s$), 8WQB ($^{un}$CRL2$^{FEM1B}$-CCDC89 dimer$^a$),

8WQC ($^{N8}$CRL2$^{FEM1B}$-CDK5R1 dimer$^{S2}$), 8WQD (FEM1B-CCDC89 complex), 8WQE ($^{un}$CRL2$^{FEM1B}$-CUX1 dimer$^s$), 8WQF ($^{un}$CRL2$^{FEM1B}$-CUX1 dimer$^a$), 8WQG ($^{N8}$CRL2$^{FEM1B}$-CCDC89 dimer$^{S2}$), 8WQH ($^{N8}$CRL2$^{FEM1B}$-CCDC89 dimer$^{S1}$), and 8WQI (FEM1B-CUX1 complex). Source data are provided with this paper.

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

## Acknowledgements

We thank Dr. Yong-Xiang Gao and the Cryo-EM Center at the University of Science and Technology of China for technical support with cryo-EM data collection; Prof. Yarden Opatowsky and Julia Guez-Haddad for gel filtration. This work is supported by the National Key R&D Program of China 2022YFA1303100, the National Natural Science Foundation of China (22137007 and 92253301), the Research Funds of Center for Advanced Interdisciplinary Science and Biomedicine of IHM (QYPY20230022), the Ministry of Science and Technology of China (2022YFC2303700 and 2022YFA1302700), the Strategic Priority Research Program of the Chinese Academy of Sciences (XDB0490000), the Center for Advanced Interdisciplinary Science and Biomedicine of IHM (QYPY20220019), and the Fundamental Research Funds for the Central Universities (WK9100000032, WK9100000044, and WK9100000027). C.X. is also supported by the Major/Innovative Program of Development Foundation of Hefei Center for Physical Science and Technology (2021HSC-CIP014); I.K. is supported by the European Research Council (ERC-2020-STG 947709), Israel Science Foundation

(ISF Grants No. 2380/21 and 3096/21), Alon Fellowship and Applebaum Foundation. R.T.T. is a Sir Henry Wellcome Postdoctoral Fellow (201387/Z/16/Z) and a Pemberton-Trinity Fellow.

## Author contributions

X.C., I.K., K.M., and C.X. conceived the project. X.C., S.L., Q.G., K.M., C.X., and H.Z. performed the structural biology and biochemistry experiments and analyzed the data; A.R. and I.K. performed the cell biology experiments and analyzed the data; J.Z. and X.Y. contributed custom reagents/analytic tools; R.T.T. and S.J.E. advised on the experimental design; X.C., I.K., K.Z., and C.X. wrote the paper with input from all the authors; I.K., K.Z., and C.X. supervised the project.

## Competing interests

S.J.E. is a founder of MAZE Therapeutics, Mirimus, TSCAN Therapeutics and ImmuneID, and serves on the scientific advisory board of Homology Medicines, TSCAN Therapeutics and MAZE Therapeutics; none of these associations impacts this work. All other authors declare no competing interests.
