## [Peer Review File · Nature Communications]

Reviewers' Comments:

Reviewer #1:

Remarks to the Author:

The manuscript describes an investigation into the large family of cullin-RING E3 ubiquitin ligases (CRLs), focusing on the subfamily containing the CUL2 subunit and the substrate receptor FEM1B. There is much buzz surrounding the CRLs these days, with their comprising perhaps 50% of all E3 ligases in humans and due to their involvement in the excited drug discovery platform of targeted protein degradation. It is fair to say that there is a very large audience of scientists who want for additional mechanistic insight into CRL mechanism and function. Hence, this study is timely.

The investigation is based on several cryo-EM structures of CRL2-FEM1B, comprising the subunits RBX1, CUL2, ELOGIN B/C, and FEM1B. The RING domain of RBX1 recruits an E2 enzyme that catalyzes ubiquitin transfer to the substrate, whereas FEM1B recognizes and binds to protein substrates by virtue of several residues located at the substrate's C-terminus. CRLs are activated in cells by the conjugation of the ubiquitin-like protein NEDD8 to the cullin subunit, and several cryo-EM structures of neddylated CRL2-FEM1B were additionally solved and analyzed. One major observation from the structural studies is that both the unmodified and neddylated CRL2-FEM1B complexes form dimers of at least two distinct conformations. Structures were also solved in the presence of peptides corresponding to the substrate proteins CCDC89 and CUX1, revealed a novel mode of binding where a His or aromatic residue located approximately 20 residues upstream from the C-terminus played important roles in high affinity binding to FEM1B. Several mutations were made in the peptide substrate, in the binding pockets of FEM1B, and in the dimerization component of FEM1B. These mutants were tested in substrate binding assays performed by isothermal calorimetry, biochemical ubiquitylation assays, and in cells using the previously described global protein stability assay (GPS).

Taken together, the paper is well-written, the experiments appeared for the most part to be performed with care, and the study should be of interest to a relatively wide audience. However, there are two major issues that the reviewer believes should be resolved before publication.

Perhaps most pressing is the observation of multiple dimeric conformations of the CRL2-FEM1B complex by cryo-EM. The authors acknowledge in the Discussion (p17) that none of their structures are compatible with E2~ubiquitin binding, raising the question of whether any of these structures are relevant to how CRL2-FEM1B ubiquitylated substrates or artifacts of structural biology. While a FEM1B mutant ('4A') was shown to disrupt dimerization (Supplementary Figure 1d) and affected ubiquitylation, numerous other interfaces are described in these various structures and yet are not characterized by mutagenesis. Ultimately, this reviewer is unconvinced that dimerization is both important for CRL2-FEM1B activity and relevant in cells. The authors should consider the following:

Does the FEM1B-ELONGIN B/C complex itself dimerize? If so, what is the affinity of the complex? For instance, the story would be more convincing if the interaction was relatively high affinity (nM range). Similarly, can the authors show that FEM1B-ELONGIN B/C dimerizes in cells? And with CUL2-RBX1? I acknowledge that these latter experiments can be tricky, but at least some attempt should be made, such as gel filtration of lysates comparing WT and 4A FEM1B.

Another major issue is the *in vitro* ubiquitylation reactions (Supplementary Figure 13). While the reviewer applauds the investigators attempt to reconstitute the biochemical activity of neddylated CRL2-FEM1B, there has been no attempt to quantify the results. Nevertheless, the authors make several bold conclusions regarding their observations. For instance, on p16, lines 5-7, is stated that "CUL2 neddylation drastically increased E3 activity." In what regard? The consumption of substrate, the formation of poly-ubiquitin chains? Even though the primary data are Western blots, some level of quantification should be used here to enable comparison of WT activity with mutants. Furthermore, the reviewer was unable to find any mention of the number of technical replicates.

Additional points

1. Can the authors please provide a control for the FEM1B 5A mutant? For instance, does it bind to C-Arg substrates as well as WT FEM1B?

2. In cases where mutations to the substrate CCDC89 show defects in ubiquitylation, it is assumed by the authors that this is evidence of weaker binding to FEM1B. However, it may also be that these mutations result in defects in catalysis. The experiments should be repeated at higher peptide concentrations that saturate FEM1B.

3. On p17, lines 17-19, it is stated that "Intriguingly, the conjugation of NEDD8 molecule, induces conversion of dimer(a) to a new class of dimer that we named dimerS2, which is the active form of CRL2FEM1B as supported by functional assays." What is the evidence that this dimer is the active form of the enzyme as opposed to dimerS1, or even more so, what is the evidence that any of the structures represent the active form of the CRL?

Reviewer #2:

Remarks to the Author:

In this paper, Chen et al., determined several cryo-EM structures of CRL2-FEM1B with and without two substrate peptides, and further investigated the specificity of C-degron recognition by FEM1B using in vitro and cellular experiments. Overall, the authors have discovered previously unknown molecular features that define the substrate specificity of FEM1B and different forms of CRL2 dimerization, potentially leading to in-depth mechanistic understanding of CRLs. However, there are several major issues in the current manuscript that need to be addressed or clarified before accepted for publication.

1. The presentation of structures and related information is not very effective. It is not informative to show diagrams like Fig. 1a and 3a. What is the point of duplicating exactly the same domain diagrams only with different colors and superscripts? The overall dimer structures in the following panels are too small and with too many colors. It may be better to enlarge the structure (use the space saved from panel a), color one protomer at a time, and use nomenclatures like EB.1, EB.2 in the labeling. It is also not very helpful to show a cartoon diagram next to the structure that is not in the same orientation (panels Fig. 1f and 1i, 3d and 3g).

Fig. 2a and 2b are also difficult to digest with very busy colors. The surface representation is not real map and is somewhat misleading. It may be better to show a cartoon representation of the model. The same problems exist in supplementary Fig. 5a, 8a, and 15.

Finally, the local potential maps in Fig. 4, supplementary Fig. 8 and 10 seem to have been carved in pymol without surrounding maps. This makes it hard to judge the quality of the map and interpret the model.

2. The ITC experiments need to be improved. The authors chose to use SUMO fused peptides for the ITC experiments, which may complicate the results. Most ITC curves are not ideal and the fitted N is way smaller or larger than that expected from a single-binding event. In such cases, the fitted Kd cannot be trusted. For example, in Fig. 1c, the cyan curve is very different from the others, but it was fitted with a Kd of 6.8 μ M, only 4 times larger than the strongest binding peptide. Would it be possible to perform the experiments using synthesized peptides instead of SUMO fusions?

The authors should also perform control experiments such as Arg/C-degron from CDK5R1 for comparison. More control experiments such as mutating the P-1 to R, L, or Q in Pro/C-degron need to be performed. This is important for the definition of degron recognized by FEM1B. Is it really a Pro/C-degron, or there is some flexibility at the very C terminus as long as the secondary site is satisfied.

The authors showed some ITC experiments in Fig. 1b and 1c, and jumped into the structures of CRL2 complexes. The substrate specificity is not mentioned until Fig. 4. Would it be better to discuss the dimer structures at first and talk about the ITC experiments and substrates altogether

in the later sections?

3. The description of the dimer structures needs to be improved. It seems that one structure has rotational symmetry, but from the tables and description the authors did not apply C2 symmetry during the reconstruction. Is there a particular reason not applying the symmetry? If it is not strictly symmetric, then calling one dimer symmetric is misleading. The second paragraph on page 7 needs to be clarified and supplemented with clearer structure figures, better with local potential.

"Plastic dimerization" is a confusing term. The authors observed two well defined dimer structures for each sample.

Page 9, Line 21: "The overall...except that the WHB domains of both CUL2 are missing". What does "missing" mean here, not well resolved or completely disappeared? A better comparison of the map at a lower threshold needs to be performed.

Page 10, Line 4: "The dimerS2 is symmetric and mediated by the two NEDD8 molecules..." This sentence is also confusing since NEDD8 is not at the dimer interface and therefore the dimerization is not mediated by NEDD8. It is likely that NEDDylation caused some structural rearrangement which led to the new dimer form.

Page 10, Line 13: "Given that the hydrophobic patch of FEM1B is involved in dimerization assembly in all solved structures." The authors actually mentioned this twice in the text. Is this true? At least from the figures shown by the authors, dimerS2 is not dimerized through FEM1B. Line 20: "Taken together, the C-terminal hydrophobic patch of FEM1B is critical for dimerization of the N8-CRL2-FEM1B complex". Is this also true for dimerS2? It is obvious that the dimer interface is not just mediated by FEM1B. The authors seemed to have over-emphasized the role of FEM1B.

Was dimerS1 neddylated given that NEDD8 was not observed in the map?

4. The substrate specificity of FEM1B needs to be further clarified. The authors made it clear that FEM1B contains a secondary binding site for the C-degron which engages F, Y, W and H. However, the necessity and sufficiency of this new binding site are not clearly defined. How far can this secondary site be away from the C-terminus?

It looks like Pro is only favored at the C-end but not required. Therefore, calling it a Pro/C-degron may not be appropriate. Q and L also work as the very C-terminal residue, but their roles were not fully tested. It is noteworthy that the ITC experiments only measure the interactions between the substrate peptide and FEM1B, which is different from a functional assay such as in vitro ubiquitination or the cell based fluorescent assay (GPS). It is a disappointment that the authors did not extensively tested various combinations of C-degron to fully elucidate the substrate specificity of FEM1B, given that they already have the assays developed, especially the GPS assay.

Page 15, Line 8, "all three mutants demonstrated impaired PolyUb". This is a vague conclusion. It looks like H-21A mutant does not affect the level of polyubiquitination, but only the rate, whereas the other two mutants affected both.

The authors made a series of strong conclusions based on the GPS assay, however, it is recommended to be more rigorous when describing the results. In many cases, partial rescue seems to be present but the authors used strong phrases such as "was not functional" (Page 16, Line24) and "unable to promote the degradation" (Page 16, Line 26) while ignoring the minor effects.

On page 16 Line 25, the authors claimed that the GPS results are consistent with E3 activity exhibited in vitro. Actually the results are not so consistent. The in vitro ubiquitination was impaired but still quite significant based on Supplementray Fig. 13. Why in the GPS assay the substrates were barely degraded? An explanation needs to be provided.

5. Other minor issues:

Abstract: there are two "overall" in the paragraph.

Page 15, Line 14: supplementary Fig. 13d should be 13e.

Page 15, Line 23: GPS needs to be defined besides the citation.

Page 18, Line 4 "...much longer sequences of the substrates are required for docking this class of C-degrees to FEM1B". This statement requires further clarification. It is not clear what the authors mean. At least the interactions with both N- and C-arms of FEM1B are not clearly shown in the paper.

Page 23, Line 18: "refined structures were by MolProbity" a verb is missing.

Fig. 6d can be barely called a model. The role of dimerization and neddylation are not shown.

Revision summary

We thank the reviewers for their positive and constructive comments and suggestions.

In the revised manuscript, we have addressed the concerns raised by the reviewers.

Most significant points addressed:

1. To assess CRL2^{FEM1B} oligomerization *in vivo*, gel filtration analysis was performed on lysates derived from cells stably expressing WT FEM1B or the dimerization-defective (4A) mutant. The data suggest the formation of higher molecular weight complexes by CRL2^{FEM1B} *in vivo*.
2. *In vitro* ubiquitination assays were quantified to compare the activities of CRL2^{FEM1B} WT and mutants.
3. CDK5R1 C-degron was used as control C-degron for *in vitro* ITC binding experiments and *in vivo* GPS assay. Sequence alignment, mutagenesis and binding experiments indicated that CDK5R1 contains the Ψ-Arg/C-degron and binds to full length FEM1B via Phe-21 and Arg-1.
4. Quality of the ITC data was improved by optimizing the experimental condition. Subsequently, additional ITC experiments were performed to elucidate the role of CCDC89 P-1 in FEM1B binding.
5. The majority of structural figures have been revised to enhance the clarity and effectiveness of the presentation of structures and associated data.

All revisions in the updated document are highlighted in red for clarity.

Reviewer #1 (Remarks to the Author):

The manuscript describes an investigation into the large family of cullin-RING E3 ubiquitin ligases (CRLs), focusing on the subfamily containing the CUL2 subunit and the substrate receptor FEM1B. There is much buzz surrounding the CRLs these days, with their comprising perhaps 50% of all E3 ligases in humans and due to their involvement in the excited drug discovery platform of targeted protein degradation. It is fair to say that there is a very large audience of scientists who want for additional mechanistic insight into CRL mechanism and function. Hence, this study is timely.

The investigation is based on several cryo-EM structures of CRL2-FEM1B, comprising the subunits RBX1, CUL2, ELOGIN B/C, and FEM1B. The RING domain of RBX1 recruits an E2 enzyme that catalyzes ubiquitin transfer to the substrate, whereas FEM1B recognizes and binds to protein substrates by virtue of several residues located at the substrate's C-terminus. CRLs are activated in cells by the conjugation of the ubiquitin-like protein NEDD8 to the cullin subunit, and several cryo-EM structures of neddylated CRL2-FEM1B were additionally solved and analyzed. One major observation from the structural studies is that both the unmodified and neddylated CRL2-FEM1B complexes form dimers of at least two distinct conformations. Structures were also solved in the presence of peptides corresponding to the substrate proteins CCDC89 and CUX1, revealed a novel mode of binding where a His or aromatic residue located approximately 20 residues upstream from the C-terminus played important roles in high affinity binding to FEM1B. Several mutations were made in the peptide substrate, in the binding pockets of FEM1B, and in the dimerization component of FEM1B. These mutants were tested in substrate binding assays performed by isothermal calorimetry, biochemical ubiquitylation assays, and in cells using the previously described global protein stability assay (GPS).

Taken together, the paper is well-written, the experiments appeared for the most part to be performed with care, and the study should be of interest to a relatively wide audience.

However, there are two major issues that the reviewer believes should be resolved before publication.

Response: We thank the reviewer for his/her positive comments on our study. His/her valuable feedback has been incorporated into a revised version of the manuscript.

Perhaps most pressing is the observation of multiple dimeric conformations of the CRL2-FEM1B complex by cryo-EM. The authors acknowledge in the Discussion (p17) that none of their structures are compatible with E2~ubiquitin binding, raising the question of whether any of these structures are relevant to how CRL2-FEM1B ubiquitylated substrates or artifacts of structural biology. While a FEM1B mutant ('4A') was shown to disrupt dimerization (Supplementary Figure 1d) and affected ubiquitylation, numerous other interfaces are described in these various structures and yet are not characterized by mutagenesis. Ultimately, this reviewer is unconvinced that dimerization is both important for CRL2-FEM1B activity and relevant in cells. The authors should consider the following:

Does the FEM1B-ELONGIN B/C complex itself dimerize? If so, what is the affinity of the complex? For instance, the story would be more convincing if the interaction was relatively high affinity (nM range). Similarly, can the authors show that FEM1B-ELONGIN B/C dimerizes in cells? And with CUL2-RBX1? I acknowledge that these latter experiments can be tricky, but at least some attempt should be made, such as gel filtration of lysates comparing WT and 4A FEM1B.

Response: We appreciate the reviewer's thoughtful suggestion. To assess the relevance of FEM1B dimerization *in vivo*, gel filtration analysis was performed on lysates derived from cells expressing either WT FEM1B or the dimerization-deficient (4A) mutant. In brief, lysates of HEK293T cell stably expressing HA-tagged FEM1B WT or 4A mutant were loaded on superose 6 10/300 column. Then, equal volumes from each fraction were loaded on SDS-PAGE, followed by western blot using HA antibody. Fig. R1 shows that compared to 4A, WT protein is eluted in a higher molecular weight (mw) fractions across all eluted samples. The highest mw pick corresponds to dimeric

complexes (~400kDa), while 4A is mainly eluted in the monomeric size complexes (~200 kDa), suggesting that WT protein forms oligomers *in vivo*, while the 4A mutation impairs oligomer formation.

Fig. R1. Gel filtration of lysates comparing WT and 4A FEM1B. **a**, Immunoblot with HA antibody of superose 6 10/300 eluted fractions from lysates of cells expressing HA-FEM1B WT and the 4A mutant. **b**, Normalized quantification of **a**, demonstrating the gel filtration peaks of the complexes corresponding to FEM1B WT and 4A mutant.

Measuring the dissociation constant for a homodimer is challenging. However, to compare the dimerization state of FEM1B *in vivo*, we performed an immunoprecipitation of cells expressing both HA-FEM1B and MYC-FEM1B, wild type or 4A mutant. As shown in Fig. R2a, WT HA-FEM1B immunoprecipitated from cell lysates co-precipitated significantly more MYC-FEM1B compared to the 4A mutant, suggesting enhanced dimerization of WT FEM1B proteins compared to the dimerization-deficient mutant *in vivo*. Furthermore, immunoprecipitated HA-FEM1B co-precipitated a greater amount of endogenous CUL2 compared to the 4A mutant (**Fig. R2b**), corroborating the association of the CRL2^{FEM1B} dimer with more CUL2 molecules as anticipated.

Due to space limitation Figures R1 and R2 were not included in the revised manuscript.

Fig. R2. Assessing FEM1B dimerization state *in vivo*. **a.** HA-FEM1B and MYC-FEM1B, WT or 4A mutant are co-expressed in HEK293T followed by immunoprecipitation (IP) with anti-HA beads. Immunoblot shows that WT HA-FEM1B co-IP more WT MYC-FEM1B compared to the IP performed in 4A mutant expressing cells. This suggests that in cells, the WT FEM1B proteins dimerize to greater extent than the mutants do. **b.** HA-FEM1B, WT or 4A mutant stably expressed in HEK293T followed by IP with anti-HA beads. Immunoblot shows that WT HA-FEM1B co-IP more endogenous CUL2 compared to the 4A mutant suggesting that FEM1B-mediated oligomerization of CUL2-FEM1B results in pulldown of more CUL2 molecules.

*Another major issue is the *in vitro* ubiquitylation reactions (Supplementary Figure 13). While the reviewer applauds the investigators attempt to reconstitute the biochemical activity of neddylated CUL2-FEM1B, there has been no attempt to quantify the results. Nevertheless, the authors make several bold conclusions regarding their observations. For instance, on p16, lines 5-7, is stated that "CUL2 neddylation drastically increased E3 activity." In what regard? The consumption of substrate, the formation of poly-ubiquitin chains? Even though the primary data are Western blots, some level of quantification should be used here to enable comparison of WT activity with mutants. Furthermore, the reviewer was unable to find any mention of the number of technical replicates.*

Response: We thank the reviewer for this comment. We revised the supplementary figure 13 by quantifying the levels of polyubiquitinated chains on substrates, catalyzed by ^{un}FEM1B, ^{N8}FEM1B and their variants. The quantification results, from triplicates, were added in revised supplementary Fig. 13b, 13d, 13f, 13h, and 13j and are presented below (**Fig. R3**).

Fig. R3. The Fraction of polyubiquitinated substrates catalyzed by unmodified and neddylated CRL2^{FEM1B} and their variants.

Additional points

1. Can the authors please provide a control for the FEM1B 5A mutant? For instance, does it bind to C-Arg substrates as well as WT FEM1B?

Response: Our previous study (PMID: 33398168) reported that the terminal 10 residues of CDK5R1 containing Arg/C-degron, is sufficient to interact with FEM1B¹⁻³⁵⁶. Intriguingly, sequence alignment of the last 23 amino acids of FEM1B peptide substrates including CDK5R1 demonstrates that CDK5R1 also contains an upstream aromatic residue, Phe (Phe-22) (Fig. R4a). As the reviewer suggested, we employed ITC binding assay to examine the FEM1B-binding affinities of the CDK5R1 C-degrons of different lengths, including a longer version (23-aa) and a shorter version (10-aa). The binding data indicate that the 23-aa C-degron binds to FEM1B > 5-fold stronger than the 10-aa one (KDs: 1.4 vs. 7.6 μM). In contrast, both C-degrons bind to the FEM1B 5A mutant with KDs in a range of 7.8-8.8 μM (Fig. R4b). The mutagenesis and binding experiments suggest that the CDK5R1 Phe-22 likely increases the FEM1B binding affinity via contacting the Ψ pocket. A complete Ψ-Arg/C-degron is required for CDK5R1 to achieve maximal FEM1B-binding affinity. The data is included in revised Fig. 4h.

Fig. R4. The sequence alignment of CDK5R1 c23mer with other C-degrons bearing peptides (left). The ITC curves for the 23-aa and the 10-aa CDK5R1 peptides binding to FEM1B-EB-EC WT and 5A mutant (right).

2. In cases where mutations to the substrate CCDC89 show defects in ubiquitylation, it is assumed by the authors that this is evidence of weaker binding to FEM1B. However, it may also be that these mutations result in defects in catalysis. The experiments should be repeated at higher peptide concentrations that saturate FEM1B.

Response: We increased the concentration of CCDC89 P-1A by 5-fold and performed the ubiquitination assay. The higher concentration of P-1A mutant partially restored the level of polyubiquitination, suggesting that the defects in ubiquitination stems from the reduced FEM1B binding affinity rather than defects in the catalytic activity (**Fig. R5**).

Fig. R5. *In vitro* ubiquitination assay to dissect the FEM1B function toward the CCDC89 C-degron. In combination with E1 and E2, ^{N8}CRL2^{FEM1B} catalyzes the polyubiquitination of His-FLAG-GFP-fusion of CCDC89 WT (1×), CCDC89 P-1A mutant (1×), and 5-fold excess of CCDC89 P-1A mutant (5×).

3. On p17, lines 17-19, it is stated that "Intriguingly, the conjugation of NEDD8

molecule, induces conversion of dimer(a) to a new class of dimer that we named dimerS2, which is the active form of CRL2FEM1B as supported by functional assays." What is the evidence that this dimer is the active form of the enzyme as opposed to dimerS1, or even more so, what is the evidence that any of the structures represent the active form of the CRL?

Response: Given that unneddylated dimer^a structure is similar to neddylated dimer^{S1} we speculated that the distinct structure of neddylated dimer^{S2} represents the active form. Based on the reviewer suggestion we revised the sentence as follows:
“The conjugation of NEDD8 disrupts the dimer^a assembly by introducing potential steric clashes, causing structural rearrangement to form a new dimeric assembly, dimer^{S2}”.

Reviewer #2 (Remarks to the Author):

In this paper, Chen et al., determined several cryo-EM structures of CRL2-FEM1B with and without two substrate peptides, and further investigated the specificity of C-degron recognition by FEM1B using in vitro and cellular experiments. Overall, the authors have discovered previously unknown molecular features that define the substrate specificity of FEM1B and different forms of CRL2 dimerization, potentially leading to in-depth mechanistic understanding of CRLs. However, there are several major issues in the current manuscript that need to be addressed or clarified before accepted for publication.

Response: We greatly appreciate the reviewer's positive feedback on our study, and fully addressed his/her comments in the revised manuscript.

1. The presentation of structures and related information is not very effective. It is not informative to show diagrams like Fig. 1a and 3a. What is the point of duplicating exactly the same domain diagrams only with different colors and superscripts? The overall dimer structures in the following panels are too small and with too many colors. It may be better to enlarge the structure (use the space saved from panel a), color one protomer at a time, and use nomenclatures like EB.1, EB.2 in the labeling. It is also not very helpful to show a cartoon diagram next to the structure that is not in the same orientation (panels Fig. 1f and 1i, 3d and 3g).

Response: We revised the figures as the reviewer suggested. In revised Fig. 1a, the five subunits, CUL2, RBX1, FEM1B, EB, and EC are shown in different colors. In revised Fig. 1b-c, the structure of CRL2^{FEM1B} dimer^s is shown in the same orientation with one protomer colored in gray as a whole and the other protomer colored by subunits. In revised Fig. 1d-e, the CRL2^{FEM1B} dimer^s are shown in the same way. In addition, we colored the two protomers in blue and cyan, respectively, to clearly indicate the dimeric assembly (**Revised Supplementary Fig. 4a and 4c**). Fig. 3 were also revised in the same way by enlarging the structures and decreasing colors. As the reviewer suggested, the nomenclatures FEM1B.p1 and FEM1B.p2 are used for the same subunit of different

protomers.

Here the revised Figure 1 is shown as follows:

Fig. R6. The structures of unmodified CRL2^{FEM1B} (unCRL2^{FEM1B}) dimer structures.

Fig. 2a and 2b are also difficult to digest with very busy colors. The surface representation is not real map and is somewhat misleading. It may be better to show a cartoon representation of the model. The same problems exist in supplementary Fig. 5a, 8a, and 15.

Response: In revised Figs. 2a and 2b, only those subunits involved in the dimeric

assembly are colored. The cartoon representation of the model is used in revised Figs. 2a and 2b, and revised supplementary Figs. 5a, 8a, and 15.

Finally, the local potential maps in Fig. 4, supplementary Fig. 8 and 10 seem to have been carved in pymol without surrounding maps. This makes it hard to judge the quality of the map and interpret the model.

Response: We used ChimeraX to generate the revised Fig. 4, and revised supplementary Figs. 8 and 10. In these revised figures, the interaction residues were redrawn in the context of full cryo-EM maps to indicate the high quality of the map and the model.

2. The ITC experiments need to be improved. The authors chose to use SUMO fused peptides for the ITC experiments, which may complicate the results. Most ITC curves are not ideal and the fitted N is way smaller or larger than that expected from a single-binding event. In such cases, the fitted Kd cannot be trusted. For example, in Fig. 1c, the cyan curve is very different from the others, but it was fitted with a Kd of 6.8 μ M, only 4 times larger than the strongest binding peptide. Would it be possible to perform the experiments using synthesized peptides instead of SUMO fusions?

Response: We thank the reviewer for pointing this out. We opted for SUMO fused peptides over synthesized peptides for two key reasons. First, certain synthesized peptides containing several hydrophobic residues exhibited low solubility, whereas SUMO-fused peptides typically demonstrate significantly higher solubility. Second, quantifying the concentration of peptides lacking tyrosine or tryptophan residues using OD₂₈₀ is challenging. Conversely, SUMO fusion proteins allow for more accurate concentration determination using OD₂₈₀ and SDS-PAGE. Notably, the SUMO fusion peptides have been successfully employed in our previous studies (PMID: 33398168, 37844242).

Our findings align with previous studies by other researchers, which also reported smaller fitted N values (PMID: 33398168, 37844242, 33398170). Although FEM1B-EB-EC complex is stable in room temperature, it requires CUL2-RBX1 to form a

complete complex. We chose to use 30-50 μM FEM1B-EB-EC for ITC experiments, although FEM1B-EB-EC might aggregate at this concentration to some extents during ITC titration. We repeated the binding experiments for BEX2 with FEM1B several times, and provided the original ITC binding curves below:

Fig. R7. The ITC binding curve for FEM1B-EB-EC binding to the BEX2 C-degron.

As an exothermic reaction, the overall heat for the BEX2-FEM1B binding curve is not substantial, however, it's crucial to recognize that the K_D value is primarily determined by the slope of the curve (variation tendency) rather than the height. The curve is fitted very well with small error ($6.8 \pm 1.0 \mu\text{M}$) (**Fig. R7**). The ITC binding experiments were performed in duplicate or triplicate with similar curves.

Based on the literature and protocols from PEAQ-ITC (Malvern Panalytical) and TA instruments, an acceptable range for the c window is $1 < c < 1000$, and the ideal C -values might fall between 5 and 500 (**PMID: 2757186**). As long as the saturation are reached after ITC titration, even for low C -values with, the measured N -values, K_D values, as well as the thermodynamic parameters are comparable for ITC binding curves with good signal-to-noise ratio (SNR) (**PMID: 14640663**), suggesting that acceptable experimental window for ITC could be extended to lower C -values in some conditions. To show the high quality of the ITC binding data, we included original ITC

curves in revised Fig. S17.

The authors should also perform control experiments such as Arg/C-degron from CDK5R1 for comparison.

Response: As suggested by the reviewer, we have used CDK5R1 peptide in *in vitro* and *in vivo* assays.

In vitro: please also refer to similar point raised by reviewer #1. Our previous study (PMID: 33398168) reported that the terminal 10 residues of CDK5R1 containing Arg/C-degron, is sufficient to interact with FEM1B¹⁻³⁵⁶. Intriguingly, sequence alignment of the last 23 amino acids of FEM1B peptide substrates including CDK5R1 demonstrates that CDK5R1 also contains an upstream aromatic residue, Phe (Phe-22) (Fig. R4a). To examine the FEM1B-binding affinities of the CDK5R1 C-degrons of different lengths, including a longer version (23-aa) and a shorter version (10-aa), we employed ITC binding assay. The binding data indicate that the 23-aa C-degron binds to FEM1B > 5-fold stronger than the 10-aa one (K_Ds: 1.4 vs. 7.6 μM). In contrast, both C-degrons bind to the FEM1B 5A mutant with K_Ds in a range of 7.8-8.8 μM (Fig. R4). The mutagenesis and binding experiments suggest that the CDK5R1 Phe-22 likely increases the FEM1B binding affinity via contacting the Ψ pocket. A complete Ψ-Arg/C-degron is required for CDK5R1 to achieve maximal FEM1B-binding affinity. The data is included in revised Fig. 4h.

CCDC89: ⁻²²KKHSLDLLSK. (6) .KLRHLS^P⁻¹
PSMB5: ⁻²³DGWIRVSSDN. (6) .KYSGST^P⁻¹
BEX2: ⁻²³--HSLRAVST. (8) .DEFCLM^P⁻¹
CUX1: ⁻²³-KFADHLHKF. (7) .AAGDLW^Q⁻¹
MCRIP1: ⁻²³-GWTIDMRRN. (7) .LSAPG^P^L⁻¹
CDK5R1: ⁻²³-VFS^ΨDLKNES. (7) .LLLGLD^R^P⁻¹

Fig. R4. The sequence alignment of CDDK5R1 c23mer with other C-degrons bearing peptides (left). The ITC curves for the 23-aa and the 10-aa CDK5R1 peptides binding to FEM1B-EB-EC WT and 5A mutant (right).

In vivo: we generated HEK293T FEM1B knockout (KO) cells using CRISPR/Cas9. We then transduced the KO cells with lentiviral constructs encoding WT or mutant FEM1B cDNA, followed by recovery and a second transduction with GFP-fused CDK5R1 C23mer. We found that the Ψ pocket mutant and the dimerization defective mutant display impaired turnover of the Arg/C-degron substrate CDK5R1 (**Fig. R8**). Thus, similar to Pro/C-degron, substrate interaction with the aromatic pocket and dimerization of CRL2^{FEM1B} are required for efficient Arg/C-degron substrate turnover. Data is presented below and included in revised **Fig. 6c**.

Fig. R8. GPS assay to validate FEM1B residues critical for degron binding and E3 complex dimerization. Stability analysis of GFP-fused CDK5R1 C23-mer bearing Arg/C-degrons in various genetic backgrounds. The GFP/DsRed ratio was used to indicate the stability of GFP-fused C-degron and was analyzed by flow cytometry. In each case, GFP-fused degron instability is rescued in cells expressing the WT but not the mutant FEM1B proteins. Control KO cells (cells transfected with sgRNA targeting AAVSI) expressing GPS substrate served as the reference for substrate instability.

More control experiments such as mutating the P-1 to R, L, or Q in Pro/C-degron need to be performed. This is important for the definition of degron recognized by FEM1B. Is it really a Pro/C-degron, or there is some flexibility at the very C terminus as long as the secondary site is satisfied.

Response: As the reviewer suggested, we performed binding experiments of FEM1B with three CCDC89 C-degron mutants, P-1R, P-1L, and P-1Q. Consistent with our structural analysis, while the P-1R mutant displayed binding affinity comparable to wild-type (KDs: 1.9 μ M vs. 2.2 μ M), P-1A, P-1Q, and P-1L mutations decreased the

FEM1B binding affinity by 3-20-fold, (KDs: 6.7-46 μ M vs. 2.2 μ M). The binding data were added in revised Fig. 4c.

As complement to the *in vitro* binding assays here, GPS experiments were performed as part of our comprehensive efforts of mapping degrons (PMID: 37735597). We found that most FEM1B peptide substrates terminated with a Pro suggesting Pro is preferred in the terminal position. Saturation mutagenesis on selected Pro terminating peptides confirmed this and showed that the most critical residues that promotes substrates instability are Pro-1 and an upstream aromatic residue (PMID: 37735597). Based on that we named this degron as Ψ -Pro/C-degron. In rare cases where terminal Pro or Arg are missing (e.g, CUX1), a penultimate hydrophobic residue might compensate.

We have added a sentence in the revised text to clarify the importance of Pro-1: “The C-terminal proline preference observed in most FEM1B C-degrons and its established role in substrate instability (PMID: 37735597) suggest that the C-terminal proline is favored.”

The authors showed some ITC experiments in Fig. 1b and 1c, and jumped into the structures of CRL2 complexes. The substrate specificity is not mentioned until Fig. 4. Would it be better to discuss the dimer structures at first and talk about the ITC experiments and substrates altogether in the later sections?

Response: Thank you for this excellent suggestion. As the reviewer suggested, we reorganized the manuscript by presenting the dimeric structures of $^{un}CRL2^{FEM1B}$ and $^{N8}CRL2^{FEM1B}$ at first followed by binding data and the substrate recognition.

3. The description of the dimer structures needs to be improved. It seems that one structure has rotational symmetry, but from the tables and description the authors did not apply C2 symmetry during the reconstruction. Is there a particular reason not applying the symmetry? If it is not strictly symmetric, then calling one dimer symmetric is misleading. The second paragraph on page 7 needs to be clarified and supplemented with clearer structure figures, better with local potential.

Response: Imposing C2 symmetry yielded a modest improvement in the core of the

dimer interface, however, this may be accompanied by exacerbated map distortion in the vicinity of the CUL2 C-termini. Other groups also imposed C1 for dimers and tetramers to get better maps (PMID: 38332366, 38332367, 37040767).

As the reviewer suggested, we added one sentence to clarify dimers of $^{un}CRL2^{FEM1B}$ as follows, “In the dimer^s, the protomer-protomer interface is symmetric. Specifically, the two FEM1B molecules, named as FEM1B.p1 and FEM1B.p2, interact with each other via a hydrophobic patch formed by the two C-terminal ankyrin repeats (ANK9-10), ...” In addition, all panels of revised Fig. 2 were redrawn using ChimeraX, in which all structure figures are shown with real cryo-EM maps.

“Plastic dimerization” is a confusing term. The authors observed two well defined dimer structures for each sample.

Response: It was changed to “... two distinct dimerization modes.”

Page 9, Line 21: “The overall...except that the WHB domains of both CUL2 are missing”. What does “missing” mean here, not well resolved or completely disappeared? A better comparison of the map at a lower threshold needs to be performed.

Response: As shown in Fig. R9, the WHB domains of both CRL2 are invisible even in lower threshold. Therefore, we changed the sentence as follows, “... WHB domains of both CUL2 are invisible, suggesting their flexible orientation”

Fig. R9. Comparison of $^{un}CRL2^{FEM1B}$ dimer^S and $^{N8}CRL2^{FEM1B}$ dimer^{S1}. The WHB domain of $^{N8}CRL2^{FEM1B}$ is circled, which is visible in $^{N8}CRL2^{FEM1B}$ dimer^{S1} but not in $^{un}CRL2^{FEM1B}$ dimer^S even at a lower threshold.

Page 10, Line 4: “The dimer^{S2} is symmetric and mediated by the two NEDD8 molecules...” This sentence is also confusing since NEDD8 is not at the dimer interface and therefore the dimerization is not mediated by NEDD8. It is likely that NEDDylation caused some structural rearrangement which led to the new dimer form.

Response: In revised Fig. 8b, we demonstrate that Ile44 of NEDD8.1 makes hydrophobic interactions with the Phe549, Val584, and Ile587 of FEM1B.2. In addition, we agree with the reviewer that neddylation disrupts the dimer^a assembly of ^{un}CRL2^{FEM1B} for introducing potential steric clashes. We added an additional sentence in the discussion section as follows-

“The conjugation of NEDD8 disrupts the dimer^a assembly by introducing potential steric clashes, causing structural rearrangement to form a new dimeric assembly, dimer^{S2}”.

Page 10, Line 13: “Given that the hydrophobic patch of FEM1B is involved in dimerization assembly in all solved structures.” The authors actually mentioned this twice in the text. Is this true? At least from the figures shown by the authors, dimer^{S2} is not dimerized through FEM1B. Line 20: “Taken together, the C-terminal hydrophobic patch of FEM1B is critical for dimerization of the N8-CRL2-FEM1B complex”. Is this also true for dimer^{S2}? It is obvious that the dimer interface is not just mediated by FEM1B. The authors seemed to have over-emphasized the role of FEM1B.

Response: Cryo-EM structures consistently reveal a hydrophobic patch on FEM1B that plays a crucial role in dimer assembly. As shown in Fig. R10, in ^{un}CRL2^{FEM1B} dimer^s, FEM1B.1 and FEM1B.2 interact through a network of hydrophobic residues, including Phe549, Val584, Ile587. Notably, in ^{un}CRL2^{FEM1B} dimer^a, the Phe549, Val584, and Ile587 of FEM1B.2 interacts with FEM1B.1, and Phe549 of FEM1B.1 also participates in the interaction with RBX1.2. The overall architecture of ^{N8}CRL2^{FEM1B} dimer^{S1} closely resembles that of ^{un}CRL2^{FEM1B} dimer^s. However, in ^{N8}CRL2^{FEM1B} dimer^{S2}, Ile44 of NEDD8.1 occupies the hydrophobic pocket formed by Phe549, Val584, and Ile587 of FEM1B.2. Due to limitations in the NEDD8.1 map quality, only its main chain was

modeled to avoid misinterpretations. In all solved structure, the hydrophobic patch is always masked to avoid being solvent exposed. Therefore, CRL2^{FEM1B} might utilize its polymeric assembly to stabilize FEM1B by masking the hydrophobic patch.

Fig. R10. The FEM1B hydrophobic patch is involved in inter-protomer interactions in different dimeric assembly.

Was dimerS1 neddylated given that NEDD8 was not observed in the map?

Response: We added the SDS-PAGE gel for unmodified and neddylated CUL2-RBX1 (^{N8}CRL2) complexes in the revised manuscript (**revised Supplementary Fig. 1b**), which clearly demonstrated successful neddylation. Subsequently, the neddylated CUL2-RBX1 was employed for N8-CRL2^{FEM1B} complex reconstitution. It's noteworthy that in dimerS1, the WHB domain of CUL2 exhibits movement away from the complex due to the presence of NEDD8. Therefore, we concluded that the absence of NEDD8 in the corresponding cryo-EM map is likely due to the intrinsic structural flexibility, potentially causing it to adopt various conformations that may not be captured during data collection.

4. The substrate specificity of FEM1B needs to be further clarified. The authors made it clear that FEM1B contains a secondary binding site for the C-degron which engages F, Y, W and H. However, the necessity and sufficiency of this new binding site are not clearly defined. How far can this secondary site be away from the C-terminus?

Response: To define the minimal distance required between the Ψ pocket and the C-terminus, we generated several variants of the CCDC89-C- with sequential deletions in the central linker region, and examined their FEM1B-binding affinities (Supplementary Table 2). Our results demonstrate that deleting up to five residues in the C-degron linker region has no significant impact on FEM1B binding. However, a deletion of seven residues decreases the binding affinity by ~3.5-fold (K_Ds: 2.2 vs. 8.0 μ M) and further deletion of two residues results in a more substantial decrease, exceeding 20-fold (K_Ds: 2.2 vs. >50 μ M) (Fig. R11). Thus, CCDC89 C-degron, can tolerate a linker length reduction down to 15 residues between the N-terminal histidine and the C-terminus without compromising its affinity for FEM1B. The data were also added in revised Fig. 4e.

CCDC89: KK**H**SLDLLSKERELNGKLRHLS**P**
 Δ 3: KK**H**SLDLLSKEREL---LRHLS**P**
 Δ 5: KK**H**SLDLLSKER-----LRHLS**P**
 Δ 7: KK**H**SLDLLSK-----LRHLS**P**
 Δ 9: KK**H**SLDLL-----LRHLS**P**

Fig. R11. Sequences of CCDC89 C-degron variants and the ITC curves for FEM1B-EB-EC binding to different C-degron variants.

It looks like Pro is only favored at the C-end but not required. Therefore, calling it a Pro/C-degron may not be appropriate. Q and L also work as the very C-terminal residue, but their roles were not fully tested. It is noteworthy that the ITC experiments only measure the interactions between the substrate peptide and FEM1B, which is different from a functional assay such as in vitro ubiquitination or the cell based fluorescent assay (GPS). It is a disappointment that the authors did not extensively tested various combinations of C-degron to fully elucidate the substrate specificity of FEM1B, given that they already have the assays developed, especially the GPS assay.

Response: We thank the reviewer for this excellent suggestion. In fact, we have addressed it in our previous work using the GPS method (PMID: 37735597). We found a preference for proline at the C-terminus of most FEM1B peptide substrates, suggesting its importance in this position. This finding was further substantiated by extensive saturation mutagenesis on selected Pro-terminated peptides. The study revealed Pro-1 and an upstream aromatic residue as the most critical determinants for substrate instability. In fact, for peptides that terminate with Pro, Q or L substitutions are not allowed and significantly stabilized the substrates. In rare cases where the C-terminus lacks a terminal Pro (or Arg) (e.g, CUX1), a penultimate hydrophobic residue might act as a compensatory factor.

We have added several sentences in revised text as follows to clarify the importance of Pro-1:

“The C-terminal proline preference observed in most FEM1B C-degrons and its established role in substrate instability (PMID: 37735597) suggest that the C-terminal proline is favored. However, it is less critical for binding, as some substrates, such as CUX1, do not end with a proline. Notably, when Pro-1 is absent, a penultimate hydrophobic residue, such as Trp-2, might compensate, as observed in the CUX1 C-degron. Therefore, we named the C-degron characterized here as Ψ -Pro/C-degron, with Ψ indicating aromatic residues and His.”

Page 15, Line 8, “all three mutants demonstrated impaired PolyUb”. This is a vague conclusion. It looks like H-21A mutant does not affect the level of polyubiquitination,

but only the rate, whereas the other two mutants affected both.

Response: In response to this comment raised by both Reviewer #1 and Reviewer #2, we have quantified the ubiquitination assay performed in triplicate. The quantified data is now presented in the revised Supplementary Fig. 13. As evident from the quantification, nearly all mutants with the exception of the FEM1B 4D mutant (dimerization mutant) displayed compromised levels of polyubiquitination (**Fig. R3**).

Fig. R3. The Fraction of polyubiquitinated substrates catalyzed by unmodified and neddylated CRL2^{FEM1B} and their variants.

The authors made a series of strong conclusions based on the GPS assay, however, it is recommended to be more rigorous when describing the results. In many cases, partial rescue seems to be present but the authors used strong phrases such as “was not functional” (Page 16, Line24) and “unable to promote the degradation” (Page 16, Line 26) while ignoring the minor effects.

Response: Text was revised as follows:

“While exhibiting slightly reduced E3 activity *in vitro* (**Supplementary Fig. 13i-j**), the 4D mutant displayed impaired *in vivo* degradation of GFP-fusion substrates harboring either Ψ-Pro/C-degrons (**Fig. 6b**) or the Arg/C-degron (**Fig. 6c**).”;

On page 16 Line 25, the authors claimed that the GPS results are consistent with E3

activity exhibited *in vitro*. Actually the results are not so consistent. The *in vitro* ubiquitination was impaired but still quite significant based on Supplementary Fig. 13. Why in the GPS assay the substrates were barely degraded? An explanation needs to be provided.

Response: We appreciate the reviewer comment on this. Here, we propose two possible explanations for these discrepancies. First, *in vivo* CUL2 might be the limiting factor with multiple substrate receptors including FEM1B likely compete for binding to the available CUL2 pool. Similarly, distinct substrates might compete for FEM1B binding. In contrast, *In vitro* assays typically employ an excess of both the E3/substrate receptors and the substrate facilitating more frequent encounters and promoting ubiquitination. This observation suggests that even a mutant with impaired binding can still weakly interact with CUL2 *in vitro*, leading to a low level of substrate ubiquitination. In contrast, the *in vivo* environment likely presents a more limited pool of CUL2 and substrate receptors. This restricted availability might significantly reduce the opportunity for such weak interactions to occur, resulting in minimal ubiquitination for the binding-deficient mutants *in vivo*. Second, *in vivo* the initial ubiquitination might be counteracted by the deubiquitinating enzymes (DUBs) that are absent in the *in vitro* reaction. Third, *in vivo* regulation by deneddylation is not present *in vitro*. Therefore, the *in vitro* assay might reflect the maximal potential activity of the E3 ligase complex, unconstrained by these negative regulators, such as CAND1 and CSN complexes. We added it in the discussion section, and also revised the sentence as follow:

“While exhibiting slightly reduced E3 activity *in vitro* (**Supplementary Fig. 13i-j**), the 4D mutant displayed impaired *in vivo* degradation of GFP-fusion substrates harboring either Ψ -Pro/C-degrons (**Fig. 6b**) or the Arg/C-degron (**Fig. 6c**).”

5. Other minor issues:

Abstract: there are two “overall” in the paragraph.

Response: It is corrected.

Page 15, Line 14: supplementary Fig. 13d should be 13e.

Response: It is corrected.

Page 15, Line 23: GPS needs to be defined besides the citation.

Response: GPS is defined as global protein stability profiling in the revised manuscript.

Page 18, Line 4 “...much longer sequences of the substrates are required for docking this class of C-degrons to FEM1B”. This statement requires further clarification. It is not clear what the authors mean. At least the interactions with both N- and C-arms of FEM1B are not clearly shown in the paper.

Response: We changed the sentence as follows, “the C-degrons of CCDC89 and CUX1 interact with both the N- and C-arms of FEM1B, suggesting that both the Ψ and Pro-binding pockets are critical for docking this class of C-degrons to FEM1B.”

As shown in Fig. R12, the C-degrons of CCDC89 and CUX1 are long enough to contact both arms of the V-shape FEM1B molecule. These two figures are included in the revised manuscript as Fig. 5a and 5f, respectively.

Fig. R12. The complex structures of the FEM1B-CCDC89 and FEM1B-CUX1.

Page 23, Line 18: “refined structures were by MolProbity” a verb is missing.

Response: It is changed to “The refined structures were evaluated by MolProbity in Phenix.”

Fig. 6d can be barely called a model. The role of dimerization and neddylation are not shown.

Response: Fig. 6d was removed from the revised manuscript. Instead, we indicate the role of CUL2 neddylation in rearranging the assembly of CRL2^{FEM1B} in revised supplementary Figure 15.

Reviewers' Comments:

Reviewer #1:

Remarks to the Author:

The authors have sufficiently addressed my concerns in the revised manuscript. I fully support publication in Nature Communications. The reviewer also strongly urges both the authors and the editor to consider adding the results shown in the rebuttal pertaining to CRL2^{FEM1B} dimerization in cells (Figures R1 and R2) to the published manuscript, as they are both convincing and address an extremely important point regarding the biological relevance of E3 dimerization.

Reviewer #2:

Remarks to the Author:

The authors have made considerable efforts to address the concerns raised during the initial round of the review process, resulting in substantial improvements to the revised manuscript in terms of readability and scientific rigor. I commend their commitment and recommend accepting the manuscript for publication.

Reviewer #1 (Remarks to the Author):

The authors have sufficiently addressed my concerns in the revised manuscript. I fully support publication in Nature Communications. The reviewer also strongly urges both the authors and the editor to consider adding the results shown in the rebuttal pertaining to CRL2^{FEM1B} dimerization in cells (Figures R1 and R2) to the published manuscript, as they are both convincing and address an extremely important point regarding the biological relevance of E3 dimerization.

Response: We thank the reviewer for his/her suggestion and added the Figures R1 and R2 as revised Supplementary Fig. 9 in the final version.

Reviewer #2 (Remarks to the Author):

The authors have made considerable efforts to address the concerns raised during the initial round of the review process, resulting in substantial improvements to the revised manuscript in terms of readability and scientific rigor. I commend their commitment and recommend accepting the manuscript for publication.

Response: We appreciate the reviewer's positive feedback on our revision.